# Efficient Low Rank Gaussian Variational Inference for Neural Networks

**Marcin B. Tomczak**
University of Cambridge
Cambridge, CB2 1PZ, UK
mbt27@cam.ac.uk

**Siddharth Swaroop**
University of Cambridge
Cambridge, CB2 1PZ, UK
ss2163@cam.ac.uk

**Richard E. Turner**
University of Cambridge
Cambridge, CB2 1PZ, UK
ret26@cam.ac.uk

## Abstract

Bayesian neural networks are enjoying a renaissance driven in part by recent advances in variational inference (VI). The most common form of VI employs a fully factorized or mean-field distribution, but this is known to suffer from several pathologies, especially as we expect posterior distributions with highly correlated parameters. Current algorithms that capture these correlations with a Gaussian approximating family are difficult to scale to large models due to computational costs and high variance of gradient updates. By using a new form of the reparametrization trick, we derive a computationally efficient algorithm for performing VI with a Gaussian family with a low-rank plus diagonal covariance structure. We scale to deep feed-forward and convolutional architectures. We find that adding low-rank terms to parametrized diagonal covariance does not improve predictive performance except on small networks, but low-rank terms added to a constant diagonal covariance improves performance on small and large-scale network architectures.

## 1 Introduction

The application of Bayesian methods to neural networks (NNs) has the potential to address many of their issues, such as enabling decision making in the low data regime [4], or capturing epistemic uncertainty [19]. However, Bayesian methods tend to be computationally expensive, making their direct application to large models like deep NNs impractical. Variational inference (VI), one of the most commonly used approximate Bayesian schemes, has seen a series of advances, enabling it to be applied to NNs [3, 12, 21]. The most common form of VI employs a mean-field (MF) Gaussian distribution as the variational posterior [21, 35]. This fully factorized posterior can take advantage of local reparametrization [21], enabling practical applications of Bayesian NNs (BNNs).

In this paper, we focus on deriving a computationally efficient algorithm for learning Gaussian variational posteriors with correlations among hidden units. Adding correlations among hidden units theoretically increases the closeness of the variational posterior to the true posterior, so we might expect better performance, although there has also been evidence favoring simpler posteriors [40, 45]. Previous attempts to add correlations either rely on additional approximations [31], or apply the naive global reparametrization trick resulting in high variance gradient updates [34, 41, 43]. This results in slow learning and poor scalability, and this has not been shown to scale beyond very small NNs.

We develop an efficient optimization scheme for learning a Gaussian posterior with a low-rank plus diagonal covariance structure within each layer of a neural network. We do this by extending the Local Reparametrization Trick [21], which was originally designed to work only with mean-field posterior distributions. We derive an algorithm that can be applied to networks with fully connected and/or convolutional layers, and scale to large NNs such as ResNets [14]. We open-source the implementation of the algorithm derived in this paper at `https://github.com/marctom/elrgvi`.

## 2 Variational inference for neural networks

We assume a standard VI problem setup for supervised learning. Given a data set $\mathcal{D} = \{(\mathbf{x}_i, \mathbf{y}_i)\}_{i=1}^N$ consisting of inputs $\mathbf{x}_i$ and corresponding targets $\mathbf{y}_i$, a neural network $f_{\boldsymbol{\theta}}$, a likelihood function $p(\mathbf{y}|f_{\boldsymbol{\theta}}(\mathbf{x}))$ and a prior $p(\boldsymbol{\theta})$, we aim to approximate the posterior distribution $p(\boldsymbol{\theta}|\mathcal{D})$. We do this by minimizing the Kullback-Leibler divergence $\mathbb{D}_{KL}\big(q(\boldsymbol{\theta}|\boldsymbol{\lambda})||p(\boldsymbol{\theta}|\mathcal{D})\big)$ to learn a parametrized distribution $q(\boldsymbol{\theta}|\boldsymbol{\lambda})$ from variational family $\mathcal{Q}$. This minimization is equivalent to maximizing the Evidence Lower Bound (ELBO) [1, 2, 16] w.r.t. variational parameters $\boldsymbol{\lambda}$:

$$\mathcal{L}(\boldsymbol{\lambda}) = \sum_{i=1}^N \mathbb{E}_{\boldsymbol{\theta} \sim q(\cdot|\boldsymbol{\lambda})} \log p\big(\mathbf{y}_i|f_{\boldsymbol{\theta}}(\mathbf{x}_i)\big) - \mathbb{D}_{KL}\big(q(\boldsymbol{\theta}|\boldsymbol{\lambda})||p(\boldsymbol{\theta})\big). \qquad (1)$$

We can then use $\boldsymbol{\lambda}^* = \operatorname{argmax}_{\boldsymbol{\lambda}} \mathcal{L}(\boldsymbol{\lambda})$ to approximate the posterior distribution $p(\boldsymbol{\theta}|\mathcal{D})$ with $q(\boldsymbol{\theta}|\boldsymbol{\lambda}^*)$. We do this at prediction time, to approximate the predictive distribution over the target $\mathbf{y}^*$ given a test input $\mathbf{x}^*$: $p(\mathbf{y}^*|\mathbf{x}^*, \mathcal{D}) \approx \int p(\mathbf{y}^*|f_{\boldsymbol{\theta}}(\mathbf{x}^*))q(\boldsymbol{\theta}|\boldsymbol{\lambda}^*)d\boldsymbol{\theta}$. When the variational family $\mathcal{Q}$ contains the true posterior distribution $p(\boldsymbol{\theta}|\mathcal{D})$, approximate inference yields $q(\boldsymbol{\theta}|\boldsymbol{\lambda}^*) = p(\boldsymbol{\theta}|\mathcal{D})$, but this setting is usually computationally intractable. The mean-field approach factorizes the variational posterior over its dimensions, $q(\boldsymbol{\theta}|\boldsymbol{\lambda}) = \prod_{d=1}^D q(\theta_d|\lambda_d)$ [2]. This leads to a computationally tractable setup, and has been extensively used in practice [3, 12, 15, 21].

We assume that the variational posterior $q(\boldsymbol{\theta}|\boldsymbol{\lambda})$ factorizes over layers $q(\boldsymbol{\theta}|\boldsymbol{\lambda}) = \prod_{l=1}^L q_l(\boldsymbol{\theta}_l|\boldsymbol{\lambda}_l)$, where $\boldsymbol{\theta}_l$ denote the parameters associated with layer $l$. When weights $\boldsymbol{\theta}$ are random we denote this by $\boldsymbol{\theta} \sim q(\cdot|\boldsymbol{\lambda})$. In the case of a linear layer, we overload notation to index elements in matrix $\boldsymbol{\theta}$ as $\theta_{ij}$, though we also use $\boldsymbol{\theta}$ to denote a vector. We denote a single convolutional or fully connected layer in a neural network as $\mathcal{F}_{\boldsymbol{\theta}_l}(\mathbf{x}_l)$, where $\mathbf{x}_l$ is the input to layer $l$. A neural network can be represented as a composition of consecutive layers and activations, $f_{\boldsymbol{\theta}} = \mathcal{F}_{\boldsymbol{\theta}_L} \circ a_{L-1} \circ \mathcal{F}_{\boldsymbol{\theta}_{L-1}}, \ldots \circ a_1 \circ \mathcal{F}_{\boldsymbol{\theta}_1}$, where $a_l$ denote nonlinearities. Therefore the weights $\boldsymbol{\theta}_l$ for each layer $\mathcal{F}_{\boldsymbol{\theta}_l}$ can be sampled sequentially and used to calculate the forward pass $\mathcal{F}_{\boldsymbol{\theta}_l}(\mathbf{x}_l)$. While naive reparametrization $\boldsymbol{\theta}_l = g(\boldsymbol{\lambda}_l, \boldsymbol{\epsilon}_l)$, e.g. $\boldsymbol{\theta}_l = \boldsymbol{\mu}_l + \boldsymbol{\sigma}_l \odot \boldsymbol{\epsilon}$ can be used to estimate $\nabla_{\boldsymbol{\lambda}_l} \mathcal{L}(\boldsymbol{\lambda})$ [3], it leads to high computational cost or high variance gradients, as we discuss next.

**Estimating $\nabla_{\boldsymbol{\lambda}} \mathcal{L}(\boldsymbol{\lambda})$ with naive reparametrization.** Estimating the gradient $\nabla_{\boldsymbol{\lambda}} \mathcal{L}(\boldsymbol{\lambda})$ requires approximating the gradient reconstruction term $\nabla_{\boldsymbol{\lambda}} \sum_{i=1}^N \mathbb{E}_{\boldsymbol{\theta} \sim q(\cdot|\boldsymbol{\lambda})} \log p\big(\mathbf{y}_i|f_{\boldsymbol{\theta}}(\mathbf{x}_i)\big)$. When the data set $\mathcal{D}$ is large, the gradients $\nabla_{\boldsymbol{\lambda}} \mathcal{L}(\boldsymbol{\lambda})$ are estimated with two sources of stochasticity: (i) subsampling the data set $\mathcal{D}$, and (ii) Monte Carlo sampling of the variational posterior $q(\boldsymbol{\theta}|\boldsymbol{\lambda})$ [43, 17].

Estimating the reconstruction term by sharing one variational sample $\boldsymbol{\theta} \sim q(\cdot|\boldsymbol{\lambda})$ among inputs $\mathbf{x}_i$ in a data batch $\mathcal{B}$ introduces correlation between the terms $\nabla_{\boldsymbol{\lambda}} \log p(\mathbf{y}_i|f_{\boldsymbol{\theta}}(\mathbf{x}_i))$. Although this does not affect the bias of the estimator, it can lead to a dramatic increase in the estimation variance [36, 21, 46]. In comparison, this correlation is zero when we have different variational samples $\boldsymbol{\theta}^{(i)} \sim q(\cdot|\boldsymbol{\lambda})$ for each input $\mathbf{x}_i$ in the data batch. When using the naive reparametrization $\boldsymbol{\theta}_l = g(\boldsymbol{\lambda}_l, \boldsymbol{\epsilon}_l)$, obtaining different variational samples $\boldsymbol{\theta}^{(i)} \sim q(\cdot|\boldsymbol{\lambda})$ for different inputs $\mathbf{x}_i$ requires storing $|\mathcal{B}|$ copies of the network (one for each input in the data batch). This results in a memory cost of an update of parameters $\boldsymbol{\lambda}_l$ of order $\mathcal{O}(N_{l,in}N_{l,out}|\mathcal{B}|)$, where $N_{l,in}$, $N_{l,out}$ denote the layers' input/output dimensions, respectively. This cost is prohibitively large for even moderately sized networks.

**The Local Reparametrization Trick (LRT).** To reduce the excessive cost of having separate variational samples per input, we can resort to local reparametrization (reparametrizing $\mathcal{F}_{\boldsymbol{\theta}_l}(\mathbf{x}_l)$ instead of $\boldsymbol{\theta}_l$) and sample the layer's outputs $\mathcal{F}_{\boldsymbol{\theta}_l}(\mathbf{x}_l)$ directly, instead of sampling weights $\boldsymbol{\theta}_l \sim q(\cdot|\boldsymbol{\lambda}_l)$. For instance, for a mean-field Gaussian posterior $q(\boldsymbol{\theta}_l|\boldsymbol{\lambda}_l)$, the LRT [21] provides the following reparametrization of the forward pass $\mathcal{F}_{\boldsymbol{\theta}_l}(\mathbf{x}_l)$ through a fully connected layer:

$$\mathcal{F}_{\boldsymbol{\theta}_l}(\mathbf{x}_l) = \mathcal{F}_{\boldsymbol{\mu}_l}(\mathbf{x}_l) + \boldsymbol{\epsilon}_l \odot \sqrt{\mathcal{F}_{\boldsymbol{\sigma}_l^2}(\mathbf{x}_l^2)}, \qquad (2)$$

where $\boldsymbol{\theta}_l \sim \mathcal{N}(\boldsymbol{\mu}_l, \mathrm{diag}[\boldsymbol{\sigma}_l^2])$, $\boldsymbol{\epsilon}_l \sim \mathcal{N}(\mathbf{0}, \mathbf{I}_l)$ and we overloaded notation by using $\mathbf{x}^2$ and $\sqrt{\mathbf{x}}$ to denote elementwise operations, and $\odot$ to denote elementwise multiplication. The reparametrization given by Equation (2) requires two forward passes $\mathcal{F}_{\boldsymbol{\mu}_l}(\mathbf{x}_l)$ and $\mathcal{F}_{\boldsymbol{\sigma}_l^2}(\mathbf{x}_l^2)$, but allocates only $\mathcal{O}(N_{l,out}|\mathcal{B}|)$ memory, compared to $\mathcal{O}(N_{l,out}N_{l,in}|\mathcal{B}|)$ for naive reparametrization. LRT provides two benefits [21]. First, sampling $\boldsymbol{\epsilon}_l$ as opposed to the large matrix $\boldsymbol{\theta}_l$ lowers the variance of estimated gradients $\nabla_{\boldsymbol{\lambda}}\mathcal{L}(\boldsymbol{\lambda})$. Second, it is now computationally feasible to sample separate variational samples $\boldsymbol{\theta}_l^{(i)} \sim q(\cdot|\boldsymbol{\lambda}_l)$ for each data input $\mathbf{x}_i$ in the batch by simply sampling the perturbation noise $\boldsymbol{\epsilon}_l$. As argued earlier, this also reduces gradient variance during training.

**Larger variational families.** Theoretically, larger variational families $\mathcal{Q}$ (such as correlations beyond simple mean-field) improve the quality of the variational approximation $q(\boldsymbol{\theta}|\boldsymbol{\lambda}^*)$, as the global maximizer $\boldsymbol{\lambda}^*$ is closer to the true posterior, with smaller $\mathbb{D}_{KL}\big(q(\boldsymbol{\theta}|\boldsymbol{\lambda}^*)||p(\boldsymbol{\theta}|\mathcal{D})\big)$. However, this poses two challenges. First, the derivation of local reparametrization in Equation (2) relies on the fact that the sum of independent Gaussian distributions follows a Gaussian distribution. Unfortunately, when considering a Gaussian posterior $q(\boldsymbol{\theta}_l|\boldsymbol{\lambda}_l)$ with non-diagonal covariance matrix, application of the same derivation does not lead to a reduction in computational complexity. To the best of the authors' knowledge, no local reparametrization other than LRT has been proposed. Second, analytically calculating $\mathbb{D}_{KL}\big(q(\boldsymbol{\theta}_l|\boldsymbol{\lambda})||p(\boldsymbol{\theta}_l)\big)$ is computationally demanding for correlated approximate posteriors. For these reasons, existing algorithms [31, 34, 41, 43] are prone to high variance updates and/or high computational cost, preventing scalability to large models.

To complicate matters further, for BNNs, the improved quality of $q(\boldsymbol{\theta}|\boldsymbol{\lambda})$ as measured by the ELBO $\mathcal{L}(\boldsymbol{\lambda})$ does not necessarily improve predictive performance of $p(\mathbf{y}^*|\mathbf{x}^*, \mathcal{D})$. In fact, some authors present evidence that employing simpler variational posteriors $q(\boldsymbol{\theta}|\boldsymbol{\lambda})$ yields better predictive performance [40, 45], while others claim the mean-field approach apparently works well enough for deep models [8, 35, 44]. Designing prior distributions over weights for neural networks ensuring that the ELBO is a reliable indicator of the quality of predictive performance is an open research problem. The evidence we gather suggests that adding complexity to the Gaussian variational posterior via correlations does not decrease the predictive performance for small networks, and as we show in the paper, can provide visible improvements. For larger networks, the message is more complicated. Next we describe how to efficiently learn correlations among units within the same layer of a neural network.

## 3 Methods

As discussed previously, it is challenging to introduce correlations among units within the same layer $l$ into Gaussian variational posteriors $q(\boldsymbol{\theta}_l|\boldsymbol{\lambda}_l)$. To overcome these difficulties, we (i) extend the local reparametrization of $\mathcal{F}_{\boldsymbol{\theta}_l}(\mathbf{x}_l)$ beyond the class of mean-field Gaussian posteriors $q(\boldsymbol{\theta}_l|\boldsymbol{\lambda}_l)$, and (ii) demonstrate an efficient way of computing complexity penalty $\mathbb{D}_{KL}\big(q(\boldsymbol{\theta}_l|\boldsymbol{\lambda})||p(\boldsymbol{\theta}_l)\big)$. Detailed derivations can be found in Supplementary Material A.2.

**Efficient forward pass with low-rank covariance.** We consider a layer-wise variational posterior of the form $q(\boldsymbol{\theta}_l|\boldsymbol{\lambda}_l) = \mathcal{N}(\boldsymbol{\theta}_l|\boldsymbol{\mu}_l, \alpha \sum_{k=1}^K \mathbf{v}_{l,k}\mathbf{v}_{l,k}^\top + \mathrm{diag}[\boldsymbol{\sigma}_l^2])$. As the lemma below shows, we can exploit the low-rank plus diagonal structure of the approximate posterior's covariance matrix to derive a local reparametrization of a forward pass $\mathcal{F}_{\boldsymbol{\theta}_l}(\mathbf{x}_l)$.

**Lemma 1.** *Let $\boldsymbol{\theta} \sim \mathcal{N}(\boldsymbol{\theta}|\boldsymbol{\mu}, \alpha \sum_{k=1}^K \mathbf{v}_k\mathbf{v}_k^\top + diag[\boldsymbol{\sigma}^2])$. The forward pass through fully connected layer $\mathcal{F}_{\boldsymbol{\theta}}(\mathbf{x})$ can be reparametrized as*

$$\mathcal{F}_{\boldsymbol{\theta}}(\mathbf{x}) = \mathcal{F}_{\boldsymbol{\mu}}(\mathbf{x}) + \sqrt{\alpha}\sum_{k=1}^K \epsilon_k \mathcal{F}_{\mathbf{v}_k}(\mathbf{x}) + \boldsymbol{\varepsilon} \odot \sqrt{\mathcal{F}_{\boldsymbol{\sigma}^2}(\mathbf{x}^2)}, \text{ where } \epsilon_k \sim \mathcal{N}(0,1), \ \boldsymbol{\varepsilon} \sim \mathcal{N}(\mathbf{0}, \mathbf{I}). \quad (3)$$

Note that Lemma 1 does not make the assumption that vectors $\mathbf{v}_k$ are pairwise orthogonal. Additionally, the noise $\epsilon_{l,k}$ can be shifted to either the layer's inputs $\mathbf{x}_l$, or the layer's outputs $\mathcal{F}_{\boldsymbol{\theta}_l}(\mathbf{x}_l)$, due to linearity of $\mathcal{F}_{\boldsymbol{\theta}_l}$. Based on the RHS

| Algorithm | Time | Memory |
|---|---|---|
| MAP | $\mathcal{O}(N_{in}N_{out}|\mathcal{B}|)$ | $\mathcal{O}(N_{out}|\mathcal{B}|)$ |
| naive mean-field | $\mathcal{O}(N_{in}N_{out}|\mathcal{B}|)$ | $\mathcal{O}(N_{in}N_{out}|\mathcal{B}|)$ |
| mean-field (LRT) | $\mathcal{O}(2N_{in}N_{out}|\mathcal{B}|)$ | $\mathcal{O}(2N_{out}|\mathcal{B}|)$ |
| naive low-rank | $\mathcal{O}(N_{in}^3N_{out}^3 + N_{in}N_{out}|\mathcal{B}|)$ | $\mathcal{O}(N_{in}N_{out}|\mathcal{B}|)$ |
| efficient low-rank | $\mathcal{O}(K^3 + (K+2)N_{in}N_{out}|\mathcal{B}|)$ | $\mathcal{O}((K+2)N_{out}|\mathcal{B}|)$ |
| full rank | $\mathcal{O}(N_{in}^3N_{out}^3 + N_{in}^2N_{out}^2|\mathcal{B}|)$ | $\mathcal{O}(N_{out}N_{in}|\mathcal{B}|)$ |

Table 1: Computational cost to update $\boldsymbol{\lambda}_l$ per layer.

of Equation (3), sampling from $\mathcal{F}_{\boldsymbol{\theta}_l}(\mathbf{x}_l)$ requires performing a standard forward pass $\mathcal{F}_{\boldsymbol{\mu}_l}(\mathbf{x}_l)$, $K$ forward passes $\mathcal{F}_{\mathbf{v}_{l,k}}(\mathbf{x}_l)$ multiplied by scalar noise $\epsilon_{l,k}$, and a forward pass $\mathcal{F}_{\boldsymbol{\sigma}_l^2}(\mathbf{x}_l^2)$ which we take the square root of and then perturb by noise $\varepsilon_l$. For convolutional layers, the diagonal part of the covariance $\text{diag}[\boldsymbol{\sigma}_l^2]$ cannot be reparametrized as the LRT cannot be applied directly [46]. However, the low-rank component $\alpha \sum_{k=1}^K \mathbf{v}_k \mathbf{v}_k^\top$ can be still reparametrized, yielding the middle term in Equation (3). We discuss convolutional layers in detail in Supplementary Material A.4.

**Computational cost.** We now describe the computational cost per layer of different algorithms. We first consider the additional memory allocation required to estimate $\nabla_{\boldsymbol{\lambda}_l} \mathcal{L}(\boldsymbol{\lambda})$ (not the cost to store $\boldsymbol{\lambda}$). As allocating memory is time-consuming, the 'memory' column can eventually be mapped into an increased running time of the algorithm. We compare the computational cost required to estimate $\nabla_{\boldsymbol{\lambda}_l} \mathcal{L}(\boldsymbol{\lambda})$ in Table 1. Naive approaches (naive mean-field and naive low-rank) require sampling reparametrized weights $\boldsymbol{\theta} \sim q(\cdot|\boldsymbol{\lambda})$ and thus incur a large memory cost, proportional to the number of parameters in that layer $N_{in}N_{out}$. The Local Reparametrization Trick (LRT) shifts sampling noise to activations for mean-field VI, reducing the memory cost by the factor of $N_{in}$, enabling practical use. Reparametrization in the RHS of Equation (3) achieves a similar reduction for a posterior with low-rank plus diagonal covariance matrix, reducing the naive memory cost proportional to $N_{in}N_{out}$ (red) to $(K+2)N_{out}$ (blue). Our method also reduces computational time in a similar way, as seen in Table 1, and described next. See Supplementary Material A.3 for more detail.

**Efficiently calculating the complexity penalty.** We now focus on efficiently calculating the divergence between the approximate posterior and prior in Equation (1), $\mathbb{D}_{KL}\big(q(\boldsymbol{\theta}|\boldsymbol{\lambda})||p(\boldsymbol{\theta})\big)$. Since both the prior $p(\boldsymbol{\theta})$ and variational posterior $q(\boldsymbol{\theta}|\boldsymbol{\lambda})$ factorize over layers, we can also factorize the KL divergence over layers, and focus on deriving an analytical expression for $\mathbb{D}_{KL}\big(q(\boldsymbol{\theta}_l|\boldsymbol{\lambda}_l)||p(\boldsymbol{\theta}_l)\big)$. The naive approach for calculating this term is dominated by the cost of calculating the determinant $|\alpha \sum_{k=1}^K \mathbf{v}_{l,k} \mathbf{v}_{l,k}^\top + \text{diag}[\boldsymbol{\sigma}_l^2]|$, with complexity $\mathcal{O}(N_{l,in}^3 N_{l,out}^3)$, for every layer $l$. This complexity is too large to be feasible for deep models with large numbers of parameters. We desire the complexity to be linear in the product $N_{l,in}N_{l,out}$. Any dependency on the product $N_{l,in}N_{l,out}$ above linear will cause the resulting algorithm to be too slow, preventing scaling to large models. To that end, we derive the following lemma, enabling computationally tractable analytic calculation of the KL divergence $\mathbb{D}_{KL}\big(q(\boldsymbol{\theta}|\boldsymbol{\lambda})||p(\boldsymbol{\theta})\big)$ between the posterior $q(\boldsymbol{\theta}|\boldsymbol{\lambda}) = \mathcal{N}(\boldsymbol{\theta}|\boldsymbol{\mu}, \alpha \sum_{k=1}^K \mathbf{v}_k \mathbf{v}_k^\top + \text{diag}[\boldsymbol{\sigma}^2])$ and an isotropic Gaussian $p(\boldsymbol{\theta}) = \mathcal{N}(\boldsymbol{\theta}|\mathbf{0}, \gamma \mathbf{I})$.

**Lemma 2.** *Let $q(\boldsymbol{\theta}|\boldsymbol{\lambda}) = \mathcal{N}(\boldsymbol{\theta}|\boldsymbol{\mu}, \alpha \sum_{k=1}^K \mathbf{v}_k \mathbf{v}_k^\top + \text{diag}[\boldsymbol{\sigma}^2])$ and $p(\boldsymbol{\theta}) = \mathcal{N}(\boldsymbol{\theta}|\mathbf{0}, \gamma \mathbf{I})$. Then the divergence $\mathbb{D}_{KL}\big(q(\boldsymbol{\theta}|\boldsymbol{\lambda})||p(\boldsymbol{\theta})\big)$ can be calculated as*

$$\mathbb{D}_{KL}\big(q(\boldsymbol{\theta}|\boldsymbol{\lambda})||p(\boldsymbol{\theta})\big) = \frac{1}{2}\Big[\sum_{d=1}^D \Big(\frac{\sigma_d^2}{\gamma} - \log \sigma_d^2\Big) + \frac{\alpha}{\gamma}\sum_{k=1}^K \|\mathbf{v}_k\|_2^2 - \Delta + \frac{1}{\gamma}\|\boldsymbol{\mu}\|_2^2 + D(\log \gamma - 1)\Big], \quad (4)$$

*where $\mathbf{V} = [\mathbf{v}_1, \mathbf{v}_2, \ldots \mathbf{v}_k]$ is a $D \times K$ matrix and $\Delta = \log |\mathbf{I}_K + \alpha \mathbf{V}^\top \text{diag}[\boldsymbol{\sigma}^2]^{-1}\mathbf{V}|$.*

Estimating the divergence term $\mathbb{D}_{KL}\big(q(\boldsymbol{\theta}_l|\boldsymbol{\lambda}_l)||p(\boldsymbol{\theta}_l)\big)$ using Equation (4) becomes dominated by calculating the log determinant $\log |\mathbf{I}_K + \alpha \mathbf{V}^\top \text{diag}[\boldsymbol{\sigma}_l^2]^{-1}\mathbf{V}|$. This is of shape $K \times K$ as opposed to $N_{l,in}N_{l,out} \times N_{l,in}N_{l,out}$, leading to a total computational cost of order $\mathcal{O}(K^3 + 4N_{l,in}N_{l,out})$, and enabling scalability to deep models. The same reduction in computational complexity follows for the gradient $\nabla_{\boldsymbol{\lambda}_l}\mathbb{D}_{KL}\big(q(\boldsymbol{\theta}|\boldsymbol{\lambda})||p(\boldsymbol{\theta})\big)$. This discussion is summarized in the 'time' column in Table 1, demonstrating the reduction from $N_{in}^3 N_{out}^3$ (red) to $K^3$ (blue).

**Theoretical properties.** Although low-rank plus diagonal approximations have previously been considered in the context of variational inference [31, 34], the theory behind the quality of the obtained variational posterior remains unclear to the best of the authors' knowledge. We now provide theoretical properties of approximate inference employing a low-rank Gaussian approximate posterior, demonstrating the difficulty in selecting the rank $K$. We derive a result providing the analytical solution for $\{\mathbf{v}_k\}$ and $\sigma^2$ when $q(\boldsymbol{\theta}|\boldsymbol{\lambda}) = \mathcal{N}(\boldsymbol{\theta}|\boldsymbol{\mu}, \sum_{k=1}^K \mathbf{v}_k \mathbf{v}_k^\top + \sigma^2 \mathbf{I})$ and the true posterior follows Gaussian distribution $p(\boldsymbol{\theta}|\mathcal{D}) = \mathcal{N}(\boldsymbol{\theta}|\boldsymbol{\mu}_p, \boldsymbol{\Sigma}_p)$. Related problems have been studied in [25, 39, 42].

**Lemma 3.** *Denote the true and approximate posterior $p(\boldsymbol{\theta}|\mathcal{D}) = \mathcal{N}(\boldsymbol{\theta}|\boldsymbol{\mu}_p, \boldsymbol{\Sigma}_p)$ and $q(\boldsymbol{\theta}|\boldsymbol{\mu}, \mathbf{V}, \sigma^2) = \mathcal{N}(\boldsymbol{\theta}|\boldsymbol{\mu}, \boldsymbol{\Sigma}_{VI} := \mathbf{V}\mathbf{V}^\top + \sigma^2 \mathbf{I})$. Assume that $\boldsymbol{\mu}_*, \mathbf{V}_*, \sigma_*^2 = argmin_{\boldsymbol{\mu}, \mathbf{V}, \sigma^2}\mathcal{L}(\boldsymbol{\mu}, \mathbf{V}, \sigma^2)$, $rank(\mathbf{V}) = K$ and $\lambda_1 \geq \lambda_2 \geq \ldots \geq \lambda_D$ are decreasing eigenvalues of the posterior covariance $\boldsymbol{\Sigma}_p$ with*

*corresponding orthonormal eigenvectors* $\mathbf{u}_1, \mathbf{u}_2, \ldots, \mathbf{u}_D$. *Then* $\mathbf{\Sigma}_{VI^*} = \sum_{1 \leq k \leq K} \lambda_k \mathbf{u}_k \mathbf{u}_k^\top + \sum_{K+1 \leq i \leq D} \sigma_*^2 \mathbf{u}_i \mathbf{u}_i^\top$ *where* $\sigma_*^2 = (\frac{1}{D-K} \sum_{k+1 \leq i \leq D} \lambda_i^{-1})^{-1}$.

Lemma 3 shows that the quality of the posterior $q(\boldsymbol{\theta}|\boldsymbol{\lambda})$ depends on the spectral properties of the true posterior covariance matrix $\mathbf{\Sigma}_p$, which are unknown in practical settings. Even in the case of a linear model $\mathbf{y} = \boldsymbol{\theta}^\top \mathbf{x}$, where the precision of posterior distribution $\mathbf{\Sigma}_p^{-1}$ is given by $\mathbf{X}\mathbf{X}^T + \sigma_{prior}^2 \mathbf{I}$, it is possible to construct examples in which any number of components $K < D - 1$ will result in large $\mathbb{D}_{KL}(q(\boldsymbol{\theta}|\boldsymbol{\lambda})||p(\boldsymbol{\theta}|\mathcal{D}))$, as we can adjust the spectrum of $\mathbf{X}\mathbf{X}^T$. Even when $p(\boldsymbol{\theta}|\mathcal{D})$ is Gaussian, it is difficult to choose a specific rank $K$, suggesting we should resort to empirical evaluation.

See Supplementary Material A.2 for a derivation of the exact expression for $\mathbb{D}_{KL}(q(\boldsymbol{\theta}|\boldsymbol{\lambda})||p(\boldsymbol{\theta}|\mathcal{D}))$, which is proportional to the gap in Jensen's inequality $\log \frac{1}{D-|S|} \sum_{k \notin S} \lambda_k^{-1} - \frac{1}{D-|S|} \sum_{k \notin S} \log \lambda_k^{-1}$, where $S$ is a set of indices of $K$ largest eigenvalues. When the dimensionality $D$ is high, in general setting $K$ to a value $K << D$ cannot be expected to reduce $\mathbb{D}_{KL}(q(\boldsymbol{\theta}|\boldsymbol{\lambda})||p(\boldsymbol{\theta}|\mathcal{D}))$ by a large margin.

When the posterior diagonal component $\sigma^2 \mathbf{I}$ is not parametrized and $\sigma^2 \approx 0$, then the form of $\mathbf{\Sigma}_{VI^*}$ remains the same as in Lemma 3. The decrease in $\mathbb{D}_{KL}(q(\boldsymbol{\theta}|\boldsymbol{\lambda})||p(\boldsymbol{\theta}|\mathcal{D}))$ by increasing $K$ by one can be well-approximated with value proportional to $\log \lambda_i$, where $\lambda_i$ denotes the largest yet non-selected eigenvalue of posterior covariance $\mathbf{\Sigma}_p$. Therefore we can expect larger benefits to introducing low-rank components compared to the previous case, where we fully parametrized the diagonal component. However, this constant diagonal would also likely lead to an overall higher $\mathbb{D}_{KL}(q(\boldsymbol{\theta}|\boldsymbol{\lambda})||p(\boldsymbol{\theta}|\mathcal{D}))$.

**Practical algorithm.** Gaussian variational posteriors with diagonal, isotropic $\mathcal{N}(\boldsymbol{\mu}_l, \sigma_l^2 \mathbf{I}_l)$, and non-isotropic covariances $\mathcal{N}(\boldsymbol{\mu}_l, \text{diag}[\boldsymbol{\sigma}^2])$ can be boosted by adding the term $\alpha \sum_{k=1}^{K} \mathbf{v}_{l,k} \mathbf{v}_{l,k}^\top$ to the covariance matrix. Lemmas 1 and 2 provide an efficient way to learn these posteriors by reparametrizing $\mathcal{F}_{\boldsymbol{\theta}_l}(\mathbf{x}_l)$. We call the resulting algorithm Efficient Low Rank Gaussian VI: ELRG-(D)-VI, where the optional D indicates a non-isotropic diagonal component $\text{diag}[\boldsymbol{\sigma}_l^2]$. As we demonstrate in Section 5, ELRG-D-VI improves over MF-VI (within reasonable optimization time) only on very small networks. Therefore we focus on evaluating ELRG-VI, where we set $\sigma^2$ to a small constant value, which performs strongly across a wide range of network sizes and architectures. While this variational posterior has lower expressibility, it also allows scaling to deep models such as ResNets.

From an algorithmic perspective, introducing low-rank terms $\alpha \sum_{k=1}^{K} \mathbf{v}_{l,k} \mathbf{v}_{l,k}^\top$ to the posterior covariance has a regularizing effect. The forward pass $\mathcal{F}_{\boldsymbol{\theta}_l}(\mathbf{x}_l)$ now has an additional stochastic term $\sqrt{\alpha} \sum_{k=1}^{K} \epsilon_k \mathcal{F}_{\mathbf{v}_{l,k}}(\mathbf{x}_l)$, making reconstruction more difficult. The gradients of the complexity penalty $\nabla_{\mathbf{v}_{l,k}} \mathbb{D}_{KL}(q(\boldsymbol{\theta}_l|\boldsymbol{\lambda}_l)||p(\boldsymbol{\theta}_l))$ will prevent $\mathbf{v}_{l,k}$ from being squeezed to $\mathbf{0}$ as they need to strike a balance between the gradient emerging from the entropy $-\log|\mathbf{I}_K + \alpha \mathbf{V}^\top \text{diag}[\boldsymbol{\sigma}_l^2]^{-1} \mathbf{V}|$, and the $L_2$ regularization from from the trace term in $\alpha \gamma^{-1} \sum_{k=1}^{K} \|\mathbf{v}_{l,k}\|_2^2$. We set hyperparameter $\alpha$ to $\frac{1}{K}$.

**Convolutional neural networks (CNNs).** As briefly discussed before, LRT cannot be applied to convolutional layers as applying the same kernel over different patches incurs correlations between elements of output. A crude approximation is to simply ignore these dependencies and sample outputs from marginal distributions. The quality of this approximation is difficult to quantify. As shown in the Supplementary Material A.4 that the low rank component $\mathcal{F}_{\mathbf{v}_k}(\mathbf{x})$, middle term in Equation (3), can be still reparametrized for convolutional layers. Thus, fixing the diagonal component $\sigma^2 \mathbf{I}$ of covariance posterior matrix with a small constant $\sigma^2$ results in negligible contribution from the diagonal term $\mathcal{F}_{\sigma^2}(\mathbf{x})$, making the reparametrization in Equation (3) good approximation for CNNs.

# 4   Related work

Naive reparamerization $\boldsymbol{\theta}_l = g(\boldsymbol{\lambda}_l, \boldsymbol{\epsilon}_l)$ was first applied to BNNs in [3]. Local reparametrization [21] is a significant improvement, but only applies to mean-field posteriors $q(\boldsymbol{\theta}|\boldsymbol{\lambda})$. The naive reparametrization for Gaussian posteriors with sparse covariance/precision is investigated in [34, 41, 43] and has been considered for deep generative models [37]. Flipout [46] uses the algebraic property $(\mathbf{W} \odot \boldsymbol{\epsilon}_1 \boldsymbol{\epsilon}_2^\top)\mathbf{x} = \boldsymbol{\epsilon}_1 \odot \mathbf{W}(\mathbf{x} \odot \boldsymbol{\epsilon}_2)$ to cheaply sample from $\{-1, 1\}$ to decorrelate shared variational samples among inputs. Flipout also applies only to mean-field variational posteriors $q(\boldsymbol{\theta}|\boldsymbol{\lambda})$. Employing a sparse low-rank structure has been considered in numerous works focusing on shallow models [25, 30, 34, 41, 42]. Some properties of low-rank approximations are studied by

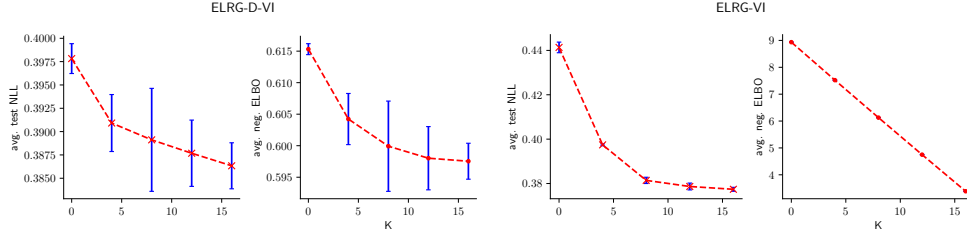

Figure 1: ELBO and test log likelihood for a small MLP network on toy data, for ELRG-D-VI (left) and ELRG-VI (right). Introducing low rank components to a posterior covariance visibly increases ELBO and test log likelihood. Error bars estimated over 30 seeds.

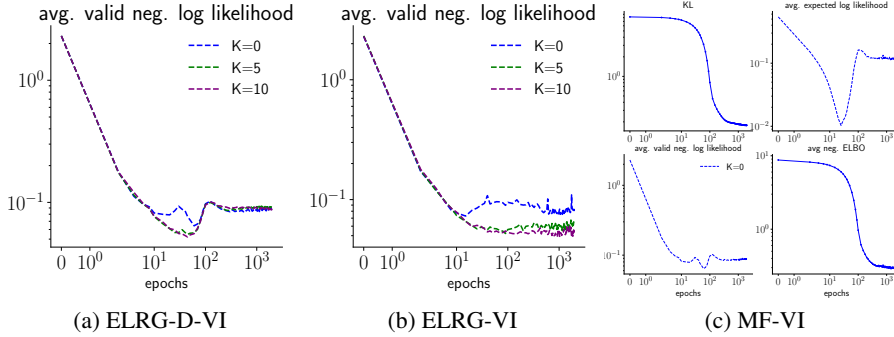

Figure 2: Adding low-rank terms to the covariance of Gaussian posteriors: (a) non-isotropic diag$[\boldsymbol{\sigma}^2]$, (b) isotropic diag$[\boldsymbol{\sigma}^2]$, (c) ordinary mean-field VI optimized with different learning rates.

[25, 38, 42]. Recently, low-rank plus diagonal covariance structure was applied to deep models with approximate VI in a natural gradient setting [31]. However, these methods only scale to small MLPs. Other approaches introducing correlated posteriors include Monte Carlo dropout [11], directional posteriors [33], and network weights defined as matrix multiplication of underlying parameters [28]. Reparametrization of models by separating the dependence between parameters and underlying noise is a well-known approach to reduce the variance of gradients [18, 22, 29, 37]. Non-reparametrized estimators of ELBO gradients for BNNs have been considered in [12, 15]. Problems with MF-VI applied to shallow BNNs are discussed in [45, 9, 10]. K-tied normal distribution [40] is another approach to alter parametrization of Gaussian variational posterior. K-tied distribution by design reduces the expressibility of variational posterior, as opposed to introducing correlations. In addition, neither mean-field Gaussian VI nor k-tied posterior can be reparametrized for convolutional layers.

## 5 Experiments

In our experiments, we focus on demonstrating the following: (i) ELRG-D-VI improves over MF-VI for small networks, but not on larger models, (ii) ELRG-VI has better predictive performance than MF-VI, (iii) ELRG-VI scales up to large CNNs and provides better predictive distributions than MAP, MF-VI and MC Dropout, (iv) sharing variational samples as in [34, 41, 43] leads to poor predictive performance. Detailed descriptions of experimental setups are in Supplementary Material A.5.

**Benefit of low-rank correlations on toy data.** We first consider learning correlated Gaussian posteriors for small MLPs. As defining priors for BNNs is a challenging task, priors selected for larger models might lead to a very low value of marginal likelihood $p(\mathcal{D})$, leading to inability to infer much information from data. For smaller MLPs, the problems with the prior assigning very low probability of data can be expected to be much smaller, making smaller models essential to study.

We consider a two dimensional synthetic classification dataset sampled from two Gaussian distributions with $\boldsymbol{\mu}_1 = [1.5, -0.5]^\top$, $\boldsymbol{\mu}_2 = [0.5, -1]^\top$ and $\boldsymbol{\Sigma}_1 = 0.25\mathbf{I}$, $\boldsymbol{\Sigma}_2 = \mathbf{I}$ using 200 data points. We compare ELRG-VI, ELRG-D-VI with MF-VI on a simple one hidden layer network, 10 hidden units per layer and ELU activations [6]. In Figure 1, we see that, as $K$ increases, the ELBO and test log

likelihood increase for both ELRG-VI and ELRG-D-VI. While ELRG-VI achieves lower value of ELBO and test log likelihood, the absolute increase of ELBO with K is constant and larger than in the case of ELRG-D-VI, similarly as in the case of the true Gaussian posterior proceeding Lemma 3.

**Larger networks.** We find that MF-VI shows underfitting on datasets, in line with previous work [45]. To show this clearly, we consider a larger network with 2 hidden layers with 100 hidden units each and ReLU activations to classify vectorized MNIST [26] images, and plot the learning curves in Figure 2. The reconstruction term and the validation negative log likelihood increase after about 100 epochs, indicating underfitting behaviour. Note that if we did not visualize the reconstruction term, but just looked at the training (negative) ELBO, this underfitting would not be obvious, and could be confused with overfitting (which would reduce the reconstruction and increase test loss [45]).

A natural question is: does adding low-rank terms to the covariance matrix improve the underfitting observed in MF-VI? We do not find that this is the case empirically. As shown in Figure 2 (a), increasing $K$ in ELRG-D-VI still leads to the underfitting phenomenon (same final negative log likelihood). The lack of improvement as $K$ increases could be because, unlike the previous toy dataset, the rank $K$ is now small compared to the rank of the covariance matrix (which is the dimension of $\boldsymbol{\theta}_l$, 10000 in this case). Unfortunately, using larger values of $K$ for larger networks is impractical as the algorithm's running time is proportional to $K$.

However, ELRG-VI, parametrized as $q(\boldsymbol{\theta}_l | \boldsymbol{\mu}_l, \{\mathbf{v}_{l,k}\}) = \mathcal{N}(\boldsymbol{\theta}_l | \boldsymbol{\mu}_l, \frac{1}{K} \sum_{k=1}^{K} \mathbf{v}_{l,k} \mathbf{v}_{l,k}^\top + \sigma_{const}^2 \mathbf{I})$, does improve predictive performance, as demonstrated in Figure 2 (b). Using $K > 0$ improves held out log likelihood by a visible margin, from $-0.076$ to $-0.057$. Based on this evidence, for larger networks we now focus on evaluating ELRG-VI, not ELRG-D-VI.

| | 400 units | | 800 units | |
|---|---|---|---|---|
| algorithm | test NLL | test error rate | test NLL | test error rate |
| ELRG-VI $K = 1$ | $-0.071 \pm 0.011$ | $1.82 \pm 0.25\%$ | $-0.070 \pm 0.014$ | $1.91 \pm 0.27\%$ |
| ELRG-VI $K = 2$ | $-0.057 \pm 0.005$ | $1.69 \pm 0.14\%$ | $\mathbf{-0.057 \pm 0.008}$ | $\mathbf{1.69 \pm 0.24\%}$ |
| ELRG-VI $K = 5$ | $\mathbf{-0.053 \pm 0.006}$ | $\mathbf{1.54 \pm 0.18\%}$ | $-0.058 \pm 0.005$ | $1.68 \pm 0.17\%$ |
| NAIVE $K = 1$ | $-0.130 \pm 0.116$ | $3.00 \pm 1.82\%$ | $-0.134 \pm 0.105$ | $2.92 \pm 1.47\%$ |
| NAIVE $K = 2$ | $-0.112 \pm 0.042$ | $2.82 \pm 0.85\%$ | $-0.113 \pm 0.028$ | $2.87 \pm 0.54\%$ |
| K-TIED $K = 2$ | $-0.105 \pm 0.004$ | $2.67 \pm 0.16\%$ | $-0.108 \pm 0.004$ | $2.61 \pm 0.17\%$ |
| K-TIED $K = 3$ | $-0.106 \pm 0.003$ | $2.69 \pm 0.15\%$ | $-0.107 \pm 0.004$ | $2.64 \pm 0.18\%$ |
| MF-VI ADAM | $-0.0964 \pm 0.001$ | $2.51 \pm 0.09\%$ | $-0.1034 \pm 0.002$ | $2.65 \pm 0.03\%$ |
| MF-VI SGD [3] | — | $1.82\%$ | — | $1.99\%$ |
| SLANG K=1 [31] | — | $2.00\%$ | — | — |
| SLANG K=32 [31] | — | $1.72\%$ | — | — |

Table 2: ELRG-VI and baselines on vectorized MNIST. ELRG-VI performs well, higher $K$ is better.

**Vectorized MNIST classification.** Previous work [3, 31] has used classification on the vectorized MNIST [26] images as a BNN benchmark, employing two hidden layer MLPs with both 400 and 800 hidden units per layer. We use the default ADAM optimizer [20], batch size of 256, 1 variational samples per update, and run optimization for 500 epochs. Results are shown in Table 2. ELRG-VI outperforms both MF-VI, naive reparametrization, k-tied Gaussian posteriors and SLANG in terms of test accuracy and test log likelihood (higher $K$ is better). Results are averaged over 10 random seeds.

**Poor performance when sharing variational samples.** We previously claimed that it is important to sample different variational samples for each input in a data batch, as this reduces gradient variance during training. To empirically test this, we compare ELRG-VI to an approach that shares a variational sample $\boldsymbol{\theta} \sim q(\cdot | \boldsymbol{\lambda})$ among inputs $\mathbf{x}_i$ on a vectorized MNIST classification task. We report the learning curves for expected and validation log likelihood for $K = 2$ and $K = 5$ in Figure 3. We observe that sharing variational samples leads to slower optimization of expected log likelihood with asymptotically suboptimal value and worse validation log likelihood by a clear margin ($-0.112$ for sharing a sample and $-0.057$ for separate samples). Additionally, the gains in performance by applying reparametrization in Lemma 3 become larger when K increases, which reveals the weakness of naive reparametrization applied to BNNs.

**LeNet image recognition.** We now investigate image classification with a simple convolutional neural network. The most recent approach introducing correlations to Gaussian posteriors [31] was not scaled beyond two fully-connected hidden layers, but we are able to scale due to our low

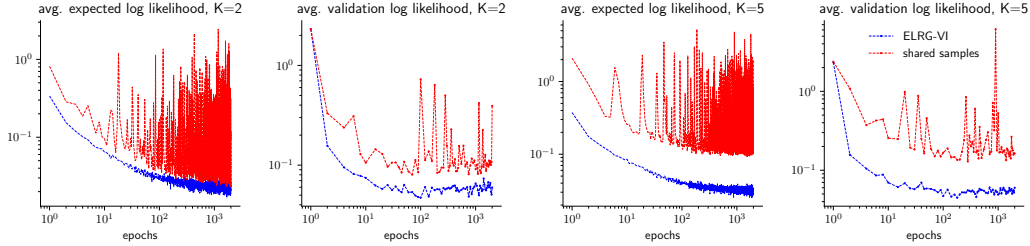

Figure 3: Comparison of the naive approach to learn low rank Gaussian variational posterior with applying the reparametrization given by Lemma 1 on a vectorized MNIST classification. Sharing variational samples $\boldsymbol{\theta} \sim q(\cdot|\boldsymbol{\lambda})$ among inputs $\mathbf{x}_i$ leads to worse performace by a clear margin.

| | neg. test log likelihood | | | | test error rate | | | |
|---|---|---|---|---|---|---|---|---|
| | MNIST | KUZUSHIJI | FASHIONMNIST | CIFAR10 | MNIST | KUZUSHIJI | FASHIONMNIST | CIFAR10 |
| ELRG-VI K=1 | $0.0274 \pm 0.002$ | $0.1714 \pm 0.004$ | $0.2609 \pm 0.006$ | $1.0955 \pm 0.005$ | $0.9 \pm 0.07\%$ | $4.75 \pm 0.09\%$ | $9.25 \pm 0.21\%$ | $33.19 \pm 0.44\%$ |
| ELRG-VI K=2 | $0.0272 \pm 0.001$ | $0.1693 \pm 0.002$ | $0.2563 \pm 0.007$ | $0.9907 \pm 0.017$ | $0.89 \pm 0.05\%$ | $4.59 \pm 0.18\%$ | $8.95 \pm 0.19\%$ | $31.80 \pm 0.30\%$ |
| ELRG-VI K=5 | $\mathbf{0.0257 \pm 0.002}$ | $\mathbf{0.1668 \pm 0.004}$ | $\mathbf{0.2522 \pm 0.004}$ | $0.8711 \pm 0.011$ | $\mathbf{0.8 \pm 0.08\%}$ | $\mathbf{4.62 \pm 0.19\%}$ | $\mathbf{9.17 \pm 0.09\%}$ | $29.43 \pm 0.32\%$ |
| MF-VI | $0.0462 \pm 0.002$ | $0.2244 \pm 0.004$ | $0.2747 \pm 0.002$ | $\mathbf{0.8579 \pm 0.003}$ | $1.09 \pm 0.08\%$ | $6.02 \pm 0.13\%$ | $9.79 \pm 0.01\%$ | $\mathbf{28.89 \pm 0.16\%}$ |
| MAP | $0.0268 \pm 0.01$ | $0.1834 \pm 0.009$ | $0.2747 \pm 0.002$ | $0.9389 \pm 0.007$ | $\mathbf{0.81 \pm 0.016\%}$ | $4.76 \pm 0.20\%$ | $9.86 \pm 0.18\%$ | $32.36 \pm 0.29\%$ |

Table 3: Results for ELRG-VI and MF-VI for image classification with LeNet CNN.

computational cost and low gradient variance during training. We experiment with common simple computer vision benchmarks: MNIST, KMNIST [5], FASHIONMNIST [47] and CIFAR10 [23]. We use the LeNet architecture [27] (two convolutional layers followed by two fully connected layers) with ReLU activations. We train all models for $500$ epochs (except MAP, which is run for $50$ epochs) using a batch size of $512$ using the ADAM optimizer [20], and do not use data augmentation. Table 3 shows log likelihoods and accuracies. ELRG-VI outperforms both MAP and MF-VI on all data sets except for CIFAR10, with significantly better predictive likelihoods and error rates. As expected, increasing rank $K$ provides better results. Interestingly, we obtained good MF-VI results on CIFAR10, far better than in reported e.g. in [35], but these results do not transfer to other datasets.

**Modern CNNs.** We now demonstrate that ELRG-VI can scale up to larger convolutional neural networks. We run ELRG-VI on RESNET [14] and ALEXNET [24], and compare with MAP estimation and MC dropout. These experiments are far larger scale than previous work with low-rank approximations [34, 41, 31]. We consider $4$ data sets: CIFAR10, CIFAR100 [23], SVHN [32] and STL10 (10 classes, 5000 images $96 \times 96$) [7]. We train all algorithms for $200$ epochs using a batch size of $256$ and the ADAM optimizer [20], with data augmentation. In Table 4, we report final test accuracy, test log likelihood (LL) and ECE score [13] to capture model calibration. We find that ELRG-VI has the best uncertainty metrics, but often worse accuracy. This is similar to the VOGN results [35]: on CIFAR10 and RESNET18, ELRG-VI $K = 4$ improves on VOGN, with accuracy of $87.24 \pm 0.37\%$ compared to $84.27\%$ and an improved negative test log likelihood of $-0.382 \pm 0.002$ compared to $-0.477$. Similarly, ELRG-VI outperforms VOGN on CIFAR10 and ALEXNET with accuracy of $76.43 \pm 0.20\%$ compared to $75.48\%$ and test log likelihood of $-0678 \pm 0.006$ vs. $-0.703$. Compared to MC Dropout, ELRG-VI yields better test log likelihood six times out of eight (they tie once), and better test ECE six times out of eight (again they tie once). MAP performs badly on uncertainty metrics (and always worse than ELRG-VI). Overall, MC Dropout performs well for the ALEXNET architecture on CIFAR10 and CIFAR100, but this is not the case for other datasets and architectures. MF-VI underfits and leads to significantly lower test log likelihood compared to other methods.

In several cases MAP estimation has good test accuracy, but at the cost of a very poor test log likelihood/calibration scores. This is a known property of MAP estimation (see e.g. Figure 3 in [13]). This is because MAP estimation aggressively picks patterns from the training data set. While this can provide better accuracy as some of the picked patterns can generalize, spurious patterns can lead to severe mistakes on the test set.

**Out of distribution test.** We plot the histogram of predictive entropies for RESNET18 trained on CIFAR100 in Figure 4. We consider test data from CIFAR100 and then out-of-distribution data from SVHN and CIFAR10. Ideally, we want lower entropy on the CIFAR100 test data, and high entropy on the out-of-distribution datasets. We observe that MAP estimation leads to confident predictions on both in distribution and out-of-distribution data, which is undesirable. On the other hand, MF-VI has under-confident predictions on both in distribution and out-of-distribution data.

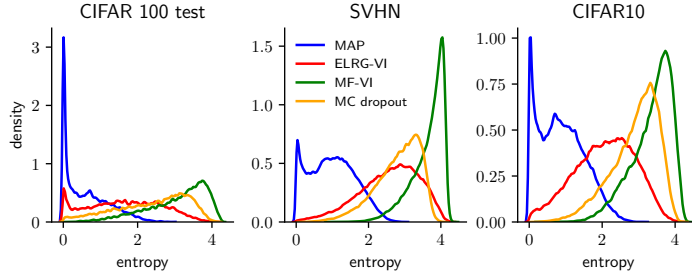

Figure 4: Predictive entropy on test data (left) and out-of-distribution data (SVHN, CIFAR10).

| RESNET18 | CIFAR10 | | | CIFAR100 | | |
|---|---|---|---|---|---|---|
| | test LL | test acc | test ECE | test LL | test acc | test ECE |
| MAP | $-0.523 \pm 0.013$ | $\mathbf{87.66 \pm 0.18}\%$ | $0.08 \pm 0.003$ | $-2.111 \pm 0.005$ | $\mathbf{59.15 \pm 0.04}\%$ | $0.218 \pm 0.001$ |
| MC Dropout | $-0.534 \pm 0.012$ | $\mathbf{87.47 \pm 0.24}\%$ | $0.084 \pm 0.001$ | $-2.121 \pm 0.02$ | $\mathbf{59.28 \pm 0.19}\%$ | $0.227 \pm 0.001$ |
| MF-VI | $-0.697 \pm 0.005$ | $76.63 \pm 0.32\%$ | $0.071 \pm 0.004$ | $-2.239 \pm 0.005$ | $41.07 \pm 0.31\%$ | $0.100 \pm 0.001$ |
| ELRG-VI $K = 1$ | $-0.413 \pm 0.004$ | $86.96 \pm 0.29\%$ | $0.044 \pm 0.003$ | $-1.819 \pm 0.021$ | $57.81 \pm 0.51\%$ | $0.154 \pm 0.005$ |
| ELRG-VI $K = 2$ | $-0.395 \pm 0.007$ | $87.09 \pm 0.21\%$ | $0.033 \pm 0.002$ | $-1.747 \pm 0.021$ | $57.85 \pm 0.60\%$ | $0.134 \pm 0.004$ |
| ELRG-VI $K = 3$ | $-0.386 \pm 0.007$ | $87.21 \pm 0.22\%$ | $0.025 \pm 0.002$ | $-1.695 \pm 0.012$ | $57.75 \pm 0.17\%$ | $0.115 \pm 0.002$ |
| ELRG-VI $K = 4$ | $\mathbf{-0.382 \pm 0.010}$ | $87.24 \pm 0.37\%$ | $\mathbf{0.018 \pm 0.002}$ | $\mathbf{-1.634 \pm 0.019}$ | $58.14 \pm 0.28\%$ | $\mathbf{0.096 \pm 0.005}$ |
| | SVHN | | | STL10 | | |
| | test LL | test acc | test ECE | test LL | test acc | test ECE |
| MAP | $-0.2 \pm 0.003$ | $95.73 \pm 0.08\%$ | $0.025 \pm 0.001$ | $-1.22 \pm 0.007$ | $\mathbf{77.21 \pm 0.06}\%$ | $0.165 \pm 0.0$ |
| MC Dropout | $-0.207 \pm 0.001$ | $95.78 \pm 0.02\%$ | $0.026 \pm 0.0$ | $-1.161 \pm 0.007$ | $76.86 \pm 0.2\%$ | $0.165 \pm 0.002$ |
| MF-VI | $-0.218 \pm 0.004$ | $94.53 \pm 0.24\%$ | $0.047 \pm 0.003$ | $-1.290 \pm 0.143$ | $71.55 \pm 1.95\%$ | $0.179 \pm 0.019$ |
| ELRG-VI $K = 1$ | $-0.158 \pm 0.002$ | $95.90 \pm 0.08\%$ | $0.010 \pm 0.001$ | $-0.811 \pm 0.018$ | $73.66 \pm 0.36\%$ | $0.080 \pm 0.012$ |
| ELRG-VI $K = 2$ | $-0.151 \pm 0.001$ | $95.95 \pm 0.02\%$ | $0.006 \pm 0.001$ | $-0.779 \pm 0.010$ | $73.23 \pm 0.36\%$ | $0.037 \pm 0.012$ |
| ELRG-VI $K = 3$ | $-0.149 \pm 0.004$ | $95.96 \pm 0.19\%$ | $0.005 \pm 0.001$ | $-0.783 \pm 0.011$ | $72.24 \pm 0.49\%$ | $0.012 \pm 0.005$ |
| ELRG-VI $K = 4$ | $\mathbf{-0.145 \pm 0.001}$ | $\mathbf{96.03 \pm 0.05}\%$ | $\mathbf{0.003 \pm 0.000}$ | $\mathbf{-0.789 \pm 0.007}$ | $72.01 \pm 0.28\%$ | $\mathbf{0.017 \pm 0.003}$ |
| ALEXNET | CIFAR10 | | | CIFAR100 | | |
| | test LL | test acc | test ECE | test LL | test acc | test ECE |
| MAP | $-0.895 \pm 0.008$ | $\mathbf{77.89 \pm 0.30}\%$ | $0.126 \pm 0.002$ | $-3.808 \pm 0.019$ | $\mathbf{46.6 \pm 0.03}\%$ | $0.326 \pm 0.003$ |
| MC Dropout | $-0.717 \pm 0.01$ | $75.22 \pm 0.328$ | $0.023 \pm 0.003$ | $\mathbf{-2.196 \pm 0.019}$ | $43.12 \pm 0.39\%$ | $\mathbf{0.022 \pm 0.002}$ |
| MF-VI | $-0.994 \pm 0.008$ | $65.38 \pm 0.32\%$ | $0.062 \pm 0.004$ | $-2.659 \pm 0.051$ | $32.41 \pm 1.39\%$ | $0.049 \pm 0.010$ |
| ELRG-VI $K = 1$ | $-0.723 \pm 0.010$ | $76.87 \pm 0.42\%$ | $0.065 \pm 0.006$ | $-2.583 \pm 0.084$ | $42.71 \pm 0.72\%$ | $0.168 \pm 0.016$ |
| ELRG-VI $K = 2$ | $-0.687 \pm 0.011$ | $76.65 \pm 0.50\%$ | $0.029 \pm 0.006$ | $-2.368 \pm 0.043$ | $42.90 \pm 0.84\%$ | $0.099 \pm 0.022$ |
| ELRG-VI $K = 3$ | $-0.681 \pm 0.007$ | $76.53 \pm 0.14\%$ | $0.021 \pm 0.004$ | $-2.275 \pm 0.021$ | $42.71 \pm 0.80\%$ | $0.074 \pm 0.014$ |
| ELRG-VI $K = 4$ | $\mathbf{-0.678 \pm 0.006}$ | $76.43 \pm 0.20\%$ | $\mathbf{0.013 \pm 0.004}$ | $-2.260 \pm 0.007$ | $42.41 \pm 0.18\%$ | $0.038 \pm 0.008$ |
| | SVHN | | | STL10 | | |
| | test LL | test acc | test ECE | test LL | test acc | test ECE |
| MAP | $\mathbf{-0.293 \pm 0.003}$ | $\mathbf{91.70 \pm 0.47}\%$ | $0.019 \pm 0.009$ | $-3.005 \pm 0.325$ | $\mathbf{65.30 \pm 0.93}\%$ | $0.138 \pm 0.053$ |
| MC Dropout | $-0.361 \pm 0.006$ | $89.70 \pm 0.15\%$ | $0.047 \pm 0.003$ | $\mathbf{-1.059 \pm 0.013}$ | $63.646 \pm 1.101$ | $0.052 \pm 0.021$ |
| MF-VI | $-0.476 \pm 0.015$ | $87.30 \pm 0.56\%$ | $0.094 \pm 0.005$ | $-1.707 \pm 0.085$ | $65.46 \pm 0.52\%$ | $0.222 \pm 0.008$ |
| ELRG-VI $K = 1$ | $-0.312 \pm 0.007$ | $90.66 \pm 0.31\%$ | $\mathbf{0.006 \pm 0.001}$ | $-1.088 \pm 0.046$ | $59.99 \pm 2.15\%$ | $\mathbf{0.018 \pm 0.008}$ |
| ELRG-VI $K = 2$ | $-0.322 \pm 0.014$ | $90.25 \pm 0.46\%$ | $0.010 \pm 0.004$ | $-1.145 \pm 0.078$ | $57.53 \pm 3.47\%$ | $\mathbf{0.018 \pm 0.002}$ |
| ELRG-VI $K = 3$ | $-0.313 \pm 0.013$ | $90.43 \pm 0.49\%$ | $0.008 \pm 0.001$ | $-1.138 \pm 0.014$ | $58.01 \pm 0.60\%$ | $0.022 \pm 0.002$ |
| ELRG-VI $K = 4$ | $-0.342 \pm 0.009$ | $89.63 \pm 0.35\%$ | $0.019 \pm 0.002$ | $-1.181 \pm 0.094$ | $56.10 \pm 4.35\%$ | $0.020 \pm 0.006$ |

Table 4: Image classification using RESNET18 and ALEXNET architectures.

We observe similar behavior for MC Dropout, in line with [35]. In comparison, ELRG-VI strikes a balance between confidence on train data, and good generalization to unseen samples.

## 6 Conclusions

We developed a scalable VI algorithm that learns Gaussian variational posteriors with non-diagonal covariance matrices by extending the Local Reparametrization Trick. Our algorithm allows us to lower the variance of ELBO gradients during training and lower the variance of the predictive distribution at test time. We added off-diagonal components to two MF-VI posteriors, first with a parametrized component, and second with a constant diagonal component. Although we did not observe improvements in performance for the former posterior when applied to larger networks, we observed that adding covariance terms to the latter posterior (ELRG-VI) prevents overfitting and provides better predictive properties. We demonstrated that ELRG-VI outperforms MF-VI in terms of predictive performance and scales to modern CNNs.

## Broader Impact

Probabilistic methods have the potential to bring considerable benefits to neural networks and computer vision [19]. These approaches return to the user multiple hypotheses that are consistent with the observed data and the user's modeling assumptions. These hypotheses can be used to carry out predictions allowing the degree of confidence in predictions to be quantified and broken apart into aleatoric contributions (e.g. label noise) and epistemic uncertainty (e.g. from lack of data). Real time modeling of epistemic uncertainty can prevent models from making catastrophic mistakes, which is crucial for real time decision making systems.

While the approach introduced in this paper is still prone to making misclassification mistakes, the model tends to output high entropy predictive distributions in this case as compared to deterministic approaches. It should be noted that the approximate inference techniques studied in this paper (Variational Inference) assumes that the data distribution is given and cannot automatically detect or remove biases existing in the data.

## Acknowledgements

Marcin B. Tomczak is supported by a Cambbridge Trust scholarship. Siddharth Swaroop is supported by an EPSRC DTP studentship. Richard E. Turner is supported by Google, Amazon, ARM, Improbable, EPSRC grants EP/M0269571 and EP/L000776/1, and the UKRI Centre for Doctoral Training in the Application of Artificial Intelligence to the study of Environmental Risks (AI4ER).

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
