[Supplementary Material · supplement.pdf]

# A Appendix: Efficient Low Rank Gaussian Variational Inference for Neural Networks

## A.1 Additional empirical results

In this section we present additional experimental results for vectorized MNIST classification.

| | 400 units | | 800 units | |
|---|---|---|---|---|
| algorithm | test NLL | test error rate | test NLL | test error rate |
| ELRG-VI $K = 1$ | $-0.071 \pm 0.011$ | $1.82 \pm 0.25\%$ | $-0.070 \pm 0.014$ | $1.91 \pm 0.27\%$ |
| ELRG-VI $K = 2$ | $-0.057 \pm 0.005$ | $1.69 \pm 0.14\%$ | $\mathbf{-0.057 \pm 0.008}$ | $\mathbf{1.69 \pm 0.24}\%$ |
| ELRG-VI $K = 3$ | $-0.055 \pm 0.005$ | $1.65 \pm 0.15\%$ | $-0.054 \pm 0.002$ | $1.63 \pm 0.12\%$ |
| ELRG-VI $K = 4$ | $-0.055 \pm 0.004$ | $1.65 \pm 0.14\%$ | $-0.054 \pm 0.004$ | $1.65 \pm 0.11\%$ |
| ELRG-VI $K = 5$ | $\mathbf{-0.053 \pm 0.006}$ | $\mathbf{1.54 \pm 0.18}\%$ | $-0.058 \pm 0.005$ | $1.68 \pm 0.17\%$ |
| NAIVE $K = 1$ | $-0.130 \pm 0.116$ | $3.00 \pm 1.82\%$ | $-0.134 \pm 0.105$ | $2.92 \pm 1.47\%$ |
| NAIVE $K = 2$ | $-0.112 \pm 0.042$ | $2.82 \pm 0.85\%$ | $-0.113 \pm 0.028$ | $2.87 \pm 0.54\%$ |
| NAIVE $K = 3$ | $-0.130 \pm 0.038$ | $3.29 \pm 0.83\%$ | $-0.454 \pm 0.984$ | $4.91 \pm 4.97\%$ |
| NAIVE $K = 4$ | $-0.163 \pm 0.060$ | $3.75 \pm 0.81\%$ | $-0.241 \pm 0.106$ | $4.66 \pm 1.23\%$ |
| K-TIED $K = 2$ | $-0.105 \pm 0.004$ | $2.67 \pm 0.16\%$ | $-0.108 \pm 0.004$ | $2.61 \pm 0.17\%$ |
| K-TIED $K = 3$ | $-0.106 \pm 0.003$ | $2.69 \pm 0.15\%$ | $-0.107 \pm 0.004$ | $2.64 \pm 0.18\%$ |
| K-TIED $K = 4$ | $-0.108 \pm 0.005$ | $2.77 \pm 0.28\%$ | $-0.109 \pm 0.002$ | $2.59 \pm 0.12\%$ |
| K-TIED $K = 5$ | $-0.104 \pm 0.005$ | $2.71 \pm 0.15\%$ | $-0.109 \pm 0.003$ | $2.63 \pm 0.09\%$ |
| MF-VI ADAM | $-0.0964 \pm 0.001$ | $2.51 \pm 0.09\%$ | $-0.1034 \pm 0.002$ | $2.65 \pm 0.03\%$ |
| MF-VI SGD [3] | $-$ | $1.82\%$ | $-$ | $1.99\%$ |
| SLANG K=1 [31] | $-$ | $2.00\%$ | $-$ | $-$ |
| SLANG K=32 [31] | $-$ | $1.72\%$ | $-$ | $-$ |

Table 5: Results for ELRG-VI and baselines on vectorized MNIST classification.

## A.2 Omitted proofs

**Lemma 1.** *Let $\boldsymbol{\theta} \sim \mathcal{N}(\boldsymbol{\theta}|\boldsymbol{\mu}, \alpha \sum_{k=1}^{K} \mathbf{v}_k \mathbf{v}_k^\top + diag[\boldsymbol{\sigma}^2])$. The forward pass through fully connected layer $\mathcal{F}_{\boldsymbol{\theta}}(\mathbf{x})$ can be reparametrized as*

$$\mathcal{F}_{\boldsymbol{\theta}}(\mathbf{x}) = \mathcal{F}_{\boldsymbol{\mu}}(\mathbf{x}) + \sqrt{\alpha} \sum_{k=1}^{K} \epsilon_k \mathcal{F}_{\mathbf{v}_k}(\mathbf{x}) + \boldsymbol{\varepsilon} \odot \sqrt{\mathcal{F}_{\boldsymbol{\sigma}^2}(\mathbf{x}^2)}, \text{ where } \epsilon_k \sim \mathcal{N}(0,1), \boldsymbol{\varepsilon} \sim \mathcal{N}(\mathbf{0},\mathbf{I}). \quad (3)$$

*Proof.* Fully connected layer. Note that when $\boldsymbol{\epsilon} \sim \mathcal{N}(\mathbf{0}, \mathbf{I}_K)$ and $\tilde{\boldsymbol{\epsilon}} \sim \mathcal{N}(\mathbf{0}, \mathbf{I}_D)$ then $\mathbf{V}\boldsymbol{\epsilon} + \text{diag}[\boldsymbol{\sigma}]\tilde{\boldsymbol{\epsilon}} \sim \mathcal{N}(\mathbf{0}, \mathbf{V}\mathbf{V}^\top + \text{diag}[\boldsymbol{\sigma}^2])$ where $V = [\mathbf{v}_1, \mathbf{v}_2, \ldots, \mathbf{v}_k]_{k=1}^K$ [42]. Also $\mathbf{V}\mathbf{V}^\top = \sum_{k=1}^K \mathbf{v}_k \mathbf{v}_k^\top$. So we have that

$$\sqrt{\alpha}\mathbf{V}\boldsymbol{\epsilon} + \text{diag}[\boldsymbol{\sigma}]\tilde{\boldsymbol{\epsilon}} = \sqrt{\alpha} \sum_{k=1}^K \epsilon_k \mathbf{v}_k + \text{diag}[\boldsymbol{\sigma}]\tilde{\boldsymbol{\epsilon}} \sim \mathcal{N}(0, \alpha \sum_{k=1}^K \mathbf{v}_k \mathbf{v}_k^\top + \text{diag}[\boldsymbol{\sigma}^2]). \quad (5)$$

We now assume $\mathbf{v}_k$ is indexed by $i$ and $j$ as it represents the layer weight matrix $\boldsymbol{\theta}$ of shape $N_{in} \times N_{out}$. We substitute sample of $\theta_{ij} = \mu_{ij} + \sqrt{\alpha} \sum_{k=1}^K \epsilon_k v_k^{ij} + \sigma_{ij}\tilde{\epsilon}_{ij}$ to obtain random variable representing layer's output $o_j$:

$$o_j = \sum_{i=1}^{N_{in}} \theta_{ij} x_i = \sum_{i=1}^{N_{in}} \sqrt{\alpha} \Big[ \mu_{ij} + \sum_{k=1}^K v_k^{ij} \epsilon_k + \sigma_{ij} \tilde{\epsilon}_{ij} \Big] x_i \quad (6)$$

$$= \sum_{i=1}^{N_{in}} \sqrt{\alpha} \sum_{k=1}^K v_k^{ij} \epsilon_k x_i + \sum_{i=1}^{N_{in}} \mu_{ij} x_i + \sigma_{ij} \tilde{\epsilon}_{ij} x_i \quad (7)$$

$$= \sqrt{\alpha} \sum_{k=1}^K \epsilon_k \sum_{i=1}^{N_{in}} v_k^{ij} x_i + \sum_{i=1}^{N_{in}} \mu_{ij} x_i + \sigma_{ij} \tilde{\epsilon}_{ij} x_i. \quad (8)$$

It follows that for the output vector $\mathbf{o}$ can be written as:

$$\mathbf{o} = \mathcal{F}_{\boldsymbol{\mu}}(\mathbf{x}) + \sqrt{\alpha} \sum_{k=1}^{K} \epsilon_k \mathcal{F}_{\boldsymbol{v}_k}(\mathbf{x}) + \mathcal{F}_{\boldsymbol{\psi}}(\mathbf{x}), \tag{9}$$

where $\boldsymbol{\psi} \sim \mathcal{N}(\mathbf{0}, \text{diag}[\boldsymbol{\sigma}^2])$. Applying Equation (2) to reparametrize the diagonal component $\mathcal{F}_{\boldsymbol{\psi}}(\mathbf{x})$ yields Equation (3). $\qquad\square$

**Lemma 2.** *Let* $q(\boldsymbol{\theta}|\boldsymbol{\lambda}) = \mathcal{N}(\boldsymbol{\theta}|\boldsymbol{\mu}, \alpha \sum_{k=1}^{K} \mathbf{v}_k \mathbf{v}_k^{\top} + \text{diag}[\boldsymbol{\sigma}^2])$ *and* $p(\boldsymbol{\theta}) = \mathcal{N}(\boldsymbol{\theta}|\mathbf{0}, \gamma\mathbf{I})$. *Then the divergence* $\mathbb{D}_{KL}\big(q(\boldsymbol{\theta}|\boldsymbol{\lambda})||p(\boldsymbol{\theta})\big)$ *can be calculated as*

$$\mathbb{D}_{KL}\big(q(\boldsymbol{\theta}|\boldsymbol{\lambda})||p(\boldsymbol{\theta})\big) = \frac{1}{2}\Big[\sum_{d=1}^{D}\Big(\frac{\sigma_d^2}{\gamma} - \log\sigma_d^2\Big) + \frac{\alpha}{\gamma}\sum_{k=1}^{K}\|\mathbf{v}_k\|_2^2 - \Delta + \frac{1}{\gamma}\|\boldsymbol{\mu}\|_2^2 + D(\log\gamma - 1)\Big], \tag{4}$$

*where* $\mathbf{V} = [\mathbf{v}_1, \mathbf{v}_2, \dots \mathbf{v}_k]$ *is a* $D \times K$ *matrix and* $\Delta = \log|\mathbf{I}_K + \alpha\mathbf{V}^{\top}\text{diag}[\boldsymbol{\sigma}^2]^{-1}\mathbf{V}|$.

*Proof.* We denote the dimension of parameters associated with the layer as $D$ and the posterior rank as $K$. Assuming that $\mathbf{v}_k$ are $D \times 1$ vectors we denote $\sum_{k=1}^{K}\mathbf{v}_k\mathbf{v}_k^{\top} = \mathbf{V}\mathbf{V}^{\top}$, where $\mathbf{V}$ is a matrix of shape $D \times K$. Let $\boldsymbol{\sigma}^2$ denote the vector of variances of shape $D \times 1$.

$$\mathbb{D}_{KL}(\mathcal{N}(\boldsymbol{\mu}, \alpha\sum_{k=1}^{K}\mathbf{v}_k\mathbf{v}_k^{\top} + \text{diag}[\boldsymbol{\sigma}^2])||\mathcal{N}(0, \gamma\mathbf{I}_D)) = \tag{10}$$

$$= \frac{1}{2}\Big[\log|\gamma\mathbf{I}| - \log\Big|\alpha\sum_{k=1}^{K}\mathbf{v}_k\mathbf{v}_k^{\top} + \text{diag}[\boldsymbol{\sigma}^2]\Big| + \frac{1}{\gamma}\text{tr}\Big[\alpha\sum_{k=1}^{K}\mathbf{v}_k\mathbf{v}_k^{\top} + \text{diag}[\boldsymbol{\sigma}^2]\Big] + \frac{\|\boldsymbol{\mu}\|_2^2}{\gamma} - D\Big] \tag{11}$$

$$= \frac{1}{2}\Big[D\log\gamma - \log\Big|\alpha\sum_{k=1}^{K}\mathbf{v}_k\mathbf{v}_k^{\top} + \text{diag}[\boldsymbol{\sigma}^2]\Big| + \frac{\alpha}{\gamma}\sum_{k=1}^{K}\|\mathbf{v}_k\|_2^2 + \frac{1}{\gamma}\sum_{i=1}^{D}\sigma_i^2 + \frac{\|\boldsymbol{\mu}\|_2^2}{\gamma} - D\Big] \tag{12}$$

$$= \frac{1}{2}\Big[D\log\gamma - \log|\alpha\mathbf{V}\mathbf{V}^{\top} + \text{diag}[\boldsymbol{\sigma}^2]| + \frac{\alpha}{\gamma}\sum_{k=1}^{K}\|\mathbf{v}_k\|_2^2 + \frac{1}{\gamma}\sum_{i=1}^{D}\sigma_i^2 + \frac{\|\boldsymbol{\mu}\|_2^2}{\gamma} - D\Big] \tag{13}$$

$$= \frac{1}{2}\Big[D(\log\gamma - 1) - \log|\text{diag}[\boldsymbol{\sigma}^2]||I_K + \alpha\mathbf{V}^{\top}\text{diag}[\boldsymbol{\sigma}^2]^{-1}\mathbf{V}| + \frac{\alpha}{\gamma}\sum_{k=1}^{K}\|\mathbf{v}_k\|_2^2 + \frac{1}{\gamma}\sum_{i=1}^{D}\sigma_i^2 + \frac{\|\boldsymbol{\mu}\|_2^2}{\gamma}\Big] \tag{14}$$

$$= \frac{1}{2}\Big[D(\log\gamma - 1) - \sum_{i=1}^{D}\log\sigma_i^2 - \log|I_K + \alpha\mathbf{V}^{\top}\text{diag}[\boldsymbol{\sigma}^2]^{-1}\mathbf{V}| + \frac{\alpha}{\gamma}\sum_{k=1}^{K}\|\mathbf{v}_k\|_2^2 + \sum_{i=1}^{D}\frac{\sigma_i^2}{\gamma} + \frac{\|\boldsymbol{\mu}\|_2^2}{\gamma}\Big] \tag{15}$$

$$= \frac{1}{2}\Big[D(\log\gamma - 1) + \sum_{i=1}^{D}\Big(\frac{\sigma_i^2}{\gamma} - \log\sigma_i^2\Big) - \log|I_K + \alpha\mathbf{V}^{\top}\text{diag}[\boldsymbol{\sigma}^2]^{-1}\mathbf{V}| + \frac{\alpha}{\gamma}\sum_{k=1}^{K}\|\mathbf{v}_k\|_2^2 + \frac{\|\boldsymbol{\mu}\|_2^2}{\gamma}\Big] \tag{16}$$

where we used Determinant Law $|\mathbf{X} + \mathbf{A}\mathbf{B}| = |\mathbf{X}||\mathbf{I} + \mathbf{B}\mathbf{X}^{-1}\mathbf{A}|$. $\qquad\square$

**Lemma 3.** *Denote the true and approximate posterior* $p(\boldsymbol{\theta}|\mathcal{D}) = \mathcal{N}(\boldsymbol{\theta}|\boldsymbol{\mu}_p, \boldsymbol{\Sigma}_p)$ *and* $q(\boldsymbol{\theta}|\boldsymbol{\mu}, \mathbf{V}, \sigma^2) = \mathcal{N}(\boldsymbol{\theta}|\boldsymbol{\mu}, \boldsymbol{\Sigma}_{VI} := \mathbf{V}\mathbf{V}^{\top} + \sigma^2\mathbf{I})$. *Assume that* $\boldsymbol{\mu}_*, \mathbf{V}_*, \sigma_*^2 = argmin_{\boldsymbol{\mu},\mathbf{V},\sigma^2}\mathcal{L}(\boldsymbol{\mu}, \mathbf{V}, \sigma^2), rank(\mathbf{V}) = K$ *and* $\lambda_1 \geq \lambda_2 \geq \dots \geq \lambda_D$ *are decreasing eigenvalues of the posterior covariance* $\boldsymbol{\Sigma}_p$ *with corresponding orthonormal eigenvectors* $\mathbf{u}_1, \mathbf{u}_2, \dots, \mathbf{u}_D$. *Then* $\boldsymbol{\Sigma}_{VI*} = \sum_{1 \leq k \leq K}\lambda_k\mathbf{u}_k\mathbf{u}_k^{\top} + \sum_{K+1 \leq i \leq D}\sigma_*^2\mathbf{u}_i\mathbf{u}_i^{\top}$ *where* $\sigma_*^2 = (\frac{1}{D-K}\sum_{K+1 \leq i \leq D}\lambda_i^{-1})^{-1}$.

We first derive helper lemmas.

**Lemma 4.** *Assume that posterior* $p(\boldsymbol{\theta}|\mathcal{D}) = \mathcal{N}(\boldsymbol{\theta}|\boldsymbol{\mu}_p, \boldsymbol{\Sigma}_p)$ *and* $q(\boldsymbol{\theta}) = \mathcal{N}(\boldsymbol{\theta}|\boldsymbol{\mu}_q, \boldsymbol{\Sigma}_q)$. *If* $\boldsymbol{\mu}^*, \boldsymbol{\Sigma}^* = argmax_{\boldsymbol{\mu}_q,\boldsymbol{\Sigma}_q}\mathcal{L}(\boldsymbol{\mu}_q, \boldsymbol{\Sigma}_q)$ *then* $\boldsymbol{\mu}^* = \boldsymbol{\mu}_p$.

*Proof.* Recall that

$$\mathbb{D}_{KL}(q(\boldsymbol{\theta}|\boldsymbol{\mu}_q, \boldsymbol{\Sigma}_q)||p(\boldsymbol{\theta}|\mathcal{D})) = \log p(\mathcal{D}) - \mathcal{L}(\boldsymbol{\mu}_q, \boldsymbol{\Sigma}_q), \tag{17}$$

and

$$\mathbb{D}_{KL}(\mathcal{N}(\boldsymbol{\mu}_1, \boldsymbol{\Sigma}_1)||\mathcal{N}(\boldsymbol{\mu}_2, \boldsymbol{\Sigma}_2)) = \frac{1}{2}\left[\log\frac{\boldsymbol{\Sigma}_2}{\boldsymbol{\Sigma}_1} + \text{tr}(\boldsymbol{\Sigma}_2^{-1}\boldsymbol{\Sigma}_1) + (\boldsymbol{\mu}_2 - \boldsymbol{\mu}_1)^T\boldsymbol{\Sigma}_2^{-1}(\boldsymbol{\mu}_2 - \boldsymbol{\mu}_1) - D\right]. \tag{18}$$

Setting $\frac{\partial\mathcal{L}(\boldsymbol{\mu}_q, \boldsymbol{\Sigma}_q)}{\partial\boldsymbol{\mu}_q}$ to 0 and solving for $\boldsymbol{\mu}_q$ gives $\boldsymbol{\mu}_q = \boldsymbol{\mu}_p$. □

We now investigate the properties of function:

$$\Phi(x_1, x_2, \ldots, x_K) := \log\frac{1}{K}\sum_{k=1}^{K}x_k - \frac{1}{K}\sum_{k=1}^{K}\log x_k, \tag{19}$$

assuming that $x_k > 0$, representing the gap in Jensen's inequality. Due to symmetry we have that $\Phi(x_1, x_2, \ldots, x_K) = \Phi(\sigma(x_1), \sigma(x_2), \ldots, \sigma(x_K))$ for any permutation $\sigma$. Thus to describe monotonicity of $\Phi$ it is enough to focus on first argument.

**Lemma 5.** $\Phi(x, x_2, \ldots, x_K)$ *is an increasing function for* $x > \frac{1}{K-1}\sum_{k=2}^{K}x_k$ *and decreasing function for* $x < \frac{1}{K-1}\sum_{k=2}^{K}x_k$.

*Proof.* Denote $\bar{x} = \frac{1}{K}\sum_{k=1}^{K}x_k$ We have that

$$\frac{\partial\Phi}{\partial x}(x, x_2, \ldots, x_k) = \frac{1}{K}\left[\frac{1}{\bar{x}} - \frac{1}{x}\right]. \tag{20}$$

It follows that

$$x > \bar{x} \iff \frac{\partial\Phi}{\partial x}(x, x_2, \ldots, x_k) > 0, \tag{21}$$

$$x < \bar{x} \iff \frac{\partial\Phi}{\partial x}(x, x_2, \ldots, x_k) < 0. \tag{22}$$

$$\tag{23}$$

Note that we have $x > \bar{x} \iff (1 - \frac{1}{K})x > \frac{1}{K}\sum_{k=2}^{K}x_k \iff \frac{K-1}{K}x > \frac{1}{K}\sum_{k=2}^{K}x_k \iff x > \frac{1}{K-1}\sum_{k=2}^{K}x_k$. Thus we obtain

$$x > \frac{1}{K-1}\sum_{k=2}^{K}x_k \iff \frac{\partial\Phi}{\partial x}(x, x_2, \ldots, x_k) > 0, \tag{24}$$

$$x < \frac{1}{K-1}\sum_{k=2}^{K}x_k \iff \frac{\partial\Phi}{\partial x}(x, x_2, \ldots, x_k) < 0. \tag{25}$$

□

**Lemma 6.** *Suppose* $x_1 < x_2 < x_3 < \ldots < x_D$ *and* $M, N$ *are* $K$ *element array of distinct increasing indices from* $\{1, 2, \ldots, D\}$. *Let* $\Phi(x_1, x_2, \ldots, x_K) = \log\frac{1}{K}\sum_{k\in M}x_k - \frac{1}{K}\sum_{k\in M}\log x_k$. *If there exists* $j \notin M$ *and* $l, r \in M$ *such that* $x_l < x_j < x_r$, *then* $\Phi(x_{M[1]}, x_{M[2]}, \ldots, x_{M[K]}) > \Phi(\arg\min_N\Phi(x_{N[1]}, x_{N[2]}, \ldots, x_{N[K]}))$. *It follows that the minimizer* $M^* = argmin_M\Phi(x_{M[1]}, x_{M[2]}, \ldots, x_{M[K]})$ *has to contain contiguous block of* $x$'s.

*Proof.* Suppose that there exists $j \notin M$ and $l, r \in M$ such that $x_l < x_j < x_r$. Assume that $x_j < \frac{1}{K-1}\sum_{k\neq j, k\in M}x_k$. Without loss of generality assume that $x_{M[1]} < x_j$, i.e. $l = M[1]$. Based on Lemma 5 it follows that $\Phi(x, x_{M[2]}, \ldots, x_{M[K]})$ is a decreasing function of $x$ for $x \in [x_{M[1]}, x_j]$. Hence replacing $M[1]$ with $j$ decreases the value of $\Phi$.

Now assume $x_j > \frac{1}{K-1}\sum_{k\neq j, k\in M}x_k$. Without loss of generality assume that $x_{M[K]} > x_j$, i.e. $r = M[K]$. Again based on Lemma 5 it follows that $\Phi(x, x_{M[2]}, \ldots, x_{M[K]})$ is an increasing function of $x$ for $x \in [x_{M[K]}, x_j]$. Hence replacing $M[K]$ with $j$ decreases the value of $\Phi$. □

*Derivation of Lemma 3.* We employ similar techniques as [42] (compare Equation (4) from [42] with the Equation below to see the difference in optimized objective). As variational posterior mean $\boldsymbol{\mu}_q$ is exact by Lemma 4 we need to minimize $\mathbb{D}_{KL}\big(\mathcal{N}(\boldsymbol{\theta}|\boldsymbol{\mu}_p, \mathbf{V}\mathbf{V}^\top + \sigma^2\mathbf{I})||\mathcal{N}(\boldsymbol{\theta}|\boldsymbol{\mu}_p, \boldsymbol{\Sigma}_p)\big)$ w.r.t. $\mathbf{V}$ and $\sigma^2$. We have that

$$\mathbb{D}_{KL}\big(\mathcal{N}(\boldsymbol{\theta}|\boldsymbol{\mu}_p, \mathbf{V}\mathbf{V}^\top + \sigma^2\mathbf{I})||\mathcal{N}(\boldsymbol{\theta}|\boldsymbol{\mu}_p, \boldsymbol{\Sigma}_p)\big) \propto -\log|\mathbf{V}\mathbf{V}^\top + \sigma^2\mathbf{I}| + \text{tr}(\boldsymbol{\Sigma}_p^{-1}(\mathbf{V}\mathbf{V}^\top + \sigma^2\mathbf{I})). \quad (26)$$

The gradient of $\mathbb{D}_{KL}\big(\mathcal{N}(\boldsymbol{\theta}|\boldsymbol{\mu}_p, \mathbf{V}\mathbf{V}^\top + \sigma^2\mathbf{I})||\mathcal{N}(\boldsymbol{\theta}|\boldsymbol{\mu}_p, \boldsymbol{\Sigma}_p)\big)$ w.r.t. $\mathbf{V}$ is:

$$\frac{\partial\mathbb{D}_{KL}\big(\mathcal{N}(\boldsymbol{\theta}|\boldsymbol{\mu}_p, \mathbf{V}\mathbf{V}^\top + \sigma^2\mathbf{I})||\mathcal{N}(\boldsymbol{\theta}|\boldsymbol{\mu}_p, \boldsymbol{\Sigma}_p)\big)}{\partial\mathbf{V}} = -(\mathbf{V}\mathbf{V}^\top + \sigma^2\mathbf{I})^{-1}\mathbf{V} + \boldsymbol{\Sigma}_p^{-1}\mathbf{V}. \quad (27)$$

Setting the above gradient to zero results in:

$$\mathbf{V} = (\mathbf{V}\mathbf{V}^\top + \sigma^2\mathbf{I})\boldsymbol{\Sigma}_p^{-1}\mathbf{V}. \quad (28)$$

We denote the precision matrix $\boldsymbol{\Lambda}_p := \boldsymbol{\Sigma}_p^{-1}$. We will denote $\lambda_k$ eigenvalues of $\boldsymbol{\Sigma}_p$ and $p_k$ eigenvalues of $\boldsymbol{\Lambda}_p$. We have that the eigenvalues of precision and covariance corresponding to the same eigenvector (assuming orthonormality) are pairwise inverses of each other $p_k = \frac{1}{\lambda_k}$.

We follow by applying SVD decomposition to $\mathbf{V}$ to obtain:

$$\mathbf{V} = \mathbf{U}\mathbf{L}\mathbf{W}^\top, \quad (29)$$

where $\mathbf{U}$ is a $d \times k$ matrix with orthonormal columns, $\mathbf{L}$ is $k \times k$ diagonal matrix and $\mathbf{W}$ is of shape $d \times k$ and also has orthonormal columns. We now have that by reexpressing Equation (28):

$$\mathbf{U}\mathbf{L}\mathbf{W}^\top = (\mathbf{U}\mathbf{L}\mathbf{W}^\top(\mathbf{U}\mathbf{L}\mathbf{W}^\top)^\top + \sigma^2\mathbf{I})\boldsymbol{\Lambda}_p\mathbf{U}\mathbf{L}\mathbf{W}^\top, \quad (30)$$

which yields:

$$\mathbf{U}\mathbf{L}\mathbf{W}^\top = (\mathbf{U}\mathbf{L}\mathbf{W}^\top\mathbf{W}\mathbf{L}\mathbf{U}^\top + \sigma^2\mathbf{I})\boldsymbol{\Lambda}_p\mathbf{U}\mathbf{L}\mathbf{W}^\top, \quad (31)$$

and:

$$\mathbf{U}\mathbf{L}\mathbf{W}^\top = (\mathbf{U}\mathbf{L}^2\mathbf{U}^\top + \sigma^2\mathbf{I})\boldsymbol{\Lambda}_p\mathbf{U}\mathbf{L}\mathbf{W}^\top, \quad (32)$$

We now multiply by $\mathbf{W}\mathbf{L}^{-1}$ on the right to obtain

$$\mathbf{U} = (\mathbf{U}\mathbf{L}^2\mathbf{U}^\top + \sigma^2\mathbf{I})\boldsymbol{\Lambda}_p\mathbf{U}. \quad (33)$$

We now analyze matrix $\mathbf{U}\mathbf{L}^2\mathbf{U}^\top + \sigma^2\mathbf{I}$. We denote $\mathbf{U} = [\mathbf{u}_1, \mathbf{u}_2, \dots, \mathbf{u}_k]$. First we note that:

$$\mathbf{U}\mathbf{L}^2\mathbf{U}^\top = \sum_{k=1}^{K} l_{kk}^2 \mathbf{u}_k\mathbf{u}_k^\top. \quad (34)$$

We also consider orthonormal compliment of $\mathbf{U}$ with basis $\tilde{\mathbf{u}}_{K-d+1}, \tilde{\mathbf{u}}_{K-D+2}, \dots, \tilde{\mathbf{u}}_D$. Note that for $n \in \{1, 2, \dots, K\}$ we have:

$$(\mathbf{U}\mathbf{L}^2\mathbf{U}^\top + \sigma^2\mathbf{I})\mathbf{u}_n = \mathbf{U}\mathbf{L}^2\mathbf{U}^\top\mathbf{u}_n + \sigma^2\mathbf{u}_n = \sum_{k=1}^{K} l_{kk}^2\mathbf{u}_k\mathbf{u}_k^\top\mathbf{u}_n + \sigma^2\mathbf{u}_n = (l_{nn}^2 + \sigma^2)\mathbf{u}_n. \quad (35)$$

We also have that for $n \in \{D - K + 1, d - K + 2, \dots, D\}$

$$(\mathbf{U}\mathbf{L}^2\mathbf{U}^\top + \sigma^2\mathbf{I})\mathbf{u}_n = \mathbf{U}\mathbf{L}^2\mathbf{U}^\top\mathbf{u}_n + \sigma^2\mathbf{u}_n = \sum_{k=1}^{K} l_{kk}^2\mathbf{u}_k\mathbf{u}_k^\top\mathbf{u}_n + \sigma^2\mathbf{u}_n = \sigma^2\mathbf{u}_n. \quad (36)$$

As $\mathbf{u}_1, \mathbf{u}_2, \dots, \mathbf{u}_K, \tilde{\mathbf{u}}_{K+1}, \tilde{\mathbf{u}}_{K+2}, \dots, \tilde{\mathbf{u}}_D$ form an orthonormal basis, we can represent the matrix $\mathbf{U}\mathbf{L}^2\mathbf{U}^\top + \sigma^2\mathbf{I}$ as:

$$\mathbf{U}\mathbf{L}^2\mathbf{U}^\top + \sigma^2\mathbf{I} = \sum_{k=1}^{K} (l_{kk}^2 + \sigma^2)\mathbf{u}_k\mathbf{u}_k^\top + \sum_{k=K+1}^{D} \sigma^2\tilde{\mathbf{u}}_\mathbf{k}\tilde{\mathbf{u}}_\mathbf{k}^\top. \quad (37)$$

Note that $\mathbf{u}_1, \mathbf{u}_2, \dots, \mathbf{u}_K, \tilde{\mathbf{u}}_{K+1}, \tilde{\mathbf{u}}_{K+2}, \dots, \tilde{\mathbf{u}}_D$ are eigenvectors of $\mathbf{U}\mathbf{L}^2\mathbf{U}^\top + \sigma^2\mathbf{I}$ by Equation (35). Thus we can derive the inverse of $\mathbf{U}\mathbf{L}^2\mathbf{U}^\top + \sigma^2\mathbf{I}$ as:

$$(\mathbf{U}\mathbf{L}^2\mathbf{U}^\top + \sigma^2\mathbf{I})^{-1} = \sum_{k=1}^{K} \frac{1}{l_{kk}^2 + \sigma^2}\mathbf{u}_k\mathbf{u}_k^\top + \sum_{k=K+1}^{D} \frac{1}{\sigma^2}\tilde{\mathbf{u}}_k\tilde{\mathbf{u}}_k^\top. \quad (38)$$

We now substitute the above equation into Equation (33) to obtain:

$$\mathbf{\Lambda}_p \mathbf{U} = \Big[ \sum_{k=1}^{K} \frac{1}{l_{kk}^2 + \sigma^2} \mathbf{u}_k \mathbf{u}_k^\top + \sum_{k=K+1}^{D} \frac{1}{\sigma^2} \tilde{\mathbf{u}}_k \tilde{\mathbf{u}}_k^\top \Big] \mathbf{U}. \tag{39}$$

Consider column $\mathbf{u}_n$ of $\mathbf{U}$. We have that:

$$\mathbf{\Lambda}_p \mathbf{u}_n = \Big[ \sum_{k=1}^{K} \frac{1}{l_{kk}^2 + \sigma^2} \mathbf{u}_k \mathbf{u}_k^\top + \sum_{k=K+1}^{D} \frac{1}{\sigma^2} \tilde{\mathbf{u}}_k \tilde{\mathbf{u}}_k^\top \Big] \mathbf{u}_n = \frac{1}{l_{nn}^2 + \sigma^2} \mathbf{u}_n. \tag{40}$$

It follows that the columns of $\mathbf{U}$, $\mathbf{u}_n$ have to be orthonormal eigenvectors of $\mathbf{\Lambda}_p$. Additionally we obtain the dependence between eigenvalues $p_k$ of $\mathbf{\Lambda}_p$ and $l_{nn}$. Based on Equation (40) we have that $p_n = \frac{1}{l_{nn}^2 + \sigma^2}$. It follows that $l_{nn}^2 = \frac{1}{p_n} - \sigma^2 = \lambda_n - \sigma^2$.

Now we consider the solutions $\mathbf{V}\mathbf{V}^\top + \sigma^2 \mathbf{I}$ which have to be of the form:

$$\mathbf{V}\mathbf{V}^\top + \sigma^2 \mathbf{I} = \mathbf{U}\mathbf{L}\mathbf{W}^\top (\mathbf{U}\mathbf{L}\mathbf{W}^\top)^\top + \sigma^2 \mathbf{I} = \mathbf{U}\mathbf{L}^2 \mathbf{U}^\top + \sigma^2 \mathbf{I}. \tag{41}$$

We eigendecompose $\mathbf{\Sigma}_p = \sum_{k=1}^{D} \lambda_k \mathbf{r}_k \mathbf{r}_k^\top$ such that $\mathbf{r}_k$ are orthonormal vectors. It follows that columns of $\mathbf{U}$ have to be in the set $\mathbf{r}_1, \mathbf{r}_2, \ldots, \mathbf{r}_d$. We denote K-element set $S$ of the selected indices of the columns. We have that

$$\mathbf{V}\mathbf{V}^\top + \sigma^2 \mathbf{I} = \sum_{k \in S} (\lambda_k - \sigma^2) \mathbf{r}_k \mathbf{r}_k^\top + \sum_{k=1}^{D} \sigma^2 \mathbf{r}_k \mathbf{r}_k^\top = \sum_{k \in S} \lambda_k \mathbf{r}_k \mathbf{r}_k^\top + \sum_{k \notin S} \sigma^2 \mathbf{r}_k \mathbf{r}_k^\top. \tag{42}$$

We now evaluate $\mathbb{D}_{KL}(\mathcal{N}(\boldsymbol{\theta}|\boldsymbol{\mu}_p, \mathbf{V}\mathbf{V}^\top + \sigma^2 \mathbf{I}) || \mathcal{N}(\boldsymbol{\theta}|\boldsymbol{\mu}_p, \mathbf{\Sigma}_p))$ as a function of $\mathbf{V}$ and $\sigma^2$ at the potential solutions $\mathbf{V}\mathbf{V}^\top + \sigma^2 \mathbf{I} = \sum_{k \in S} \lambda_k \mathbf{r}_k \mathbf{r}_k^\top + \sum_{k \notin S} \sigma^2 \mathbf{r}_k \mathbf{r}_k^\top$:

$$\mathbb{D}_{KL}(\mathcal{N}(\boldsymbol{\theta}|\boldsymbol{\mu}_p, \mathbf{V}\mathbf{V}^\top + \sigma^2 \mathbf{I}) || \mathcal{N}(\boldsymbol{\theta}|\boldsymbol{\mu}_p, \mathbf{\Sigma}_p))$$
$$\propto \log|\mathbf{\Sigma}_p| - \log\Big|\sum_{k \in S} \lambda_k \mathbf{r}_k \mathbf{r}_k^\top + \sum_{k \notin S} \sigma^2 \mathbf{r}_k \mathbf{r}_k^\top\Big| + \mathrm{tr}(\mathbf{\Lambda}_p (\sum_{k \in S} \lambda_k \mathbf{r}_k \mathbf{r}_k^\top + \sum_{k \notin S} \sigma^2 \mathbf{r}_k \mathbf{r}_k^\top)). \tag{43}$$

Since $\log\big|\sum_{k \in S} \lambda_k \mathbf{r}_k \mathbf{r}_k^\top + \sum_{k \notin S} \sigma^2 \mathbf{r}_k \mathbf{r}_k^\top\big| = \sum_{k \in S} \log \lambda_k + (D - |S|) \log \sigma^2$ it follows that

$$2\mathbb{D}_{KL}(\boldsymbol{\theta}|\mathcal{N}(\boldsymbol{\mu}_p, \mathbf{V}\mathbf{V}^\top + \sigma^2 \mathbf{I}) || \mathcal{N}(\boldsymbol{\theta}|\boldsymbol{\mu}_p, \mathbf{\Sigma}_p)) =$$

$$= \log|\mathbf{\Sigma}_p| - \sum_{k \in S} \log \lambda_k - (D - |S|) \log \sigma^2 + \mathrm{tr}(\mathbf{\Lambda}_p (\sum_{k \in S} \lambda_k \mathbf{r}_k \mathbf{r}_k^\top + \sum_{k \notin S} \sigma^2 \mathbf{r}_k \mathbf{r}_k^\top)) - D \tag{44}$$

$$= \log|\mathbf{\Sigma}_p| - \sum_{k \in S} \log \lambda_k - (D - |S|) \log \sigma^2 + \mathrm{tr}(\sum_{k \in S} \lambda_k \mathbf{\Lambda}_p \mathbf{r}_k \mathbf{r}_k^\top + \sum_{k \notin S} \sigma^2 \mathbf{\Lambda}_p \mathbf{r}_k \mathbf{r}_k^T) - D \tag{45}$$

$$= \log|\mathbf{\Sigma}_p| - \sum_{k \in S} \log \lambda_k - (D - |S|) \log \sigma^2 + \sum_{k \in S} \mathrm{tr}(\lambda_k p_k \mathbf{r}_k \mathbf{r}_k^\top) + \sum_{k \notin S} \mathrm{tr}(\sigma^2 p_k \mathbf{r}_k \mathbf{r}_k^\top) - D \tag{46}$$

$$= \log|\mathbf{\Sigma}_p| - \sum_{k \in S} \log \lambda_k - (D - |S|) \log \sigma^2 + \sum_{k \in S} \mathrm{tr}(\mathbf{r}_k^\top \mathbf{r}_k) + \sum_{k \notin S} \sigma^2 p_k \mathrm{tr}(\mathbf{r}_k^\top \mathbf{r}_k) - D \tag{47}$$

$$= \log|\mathbf{\Sigma}_p| - \sum_{k \in S} \log \lambda_k - (D - |S|) \log \sigma^2 + |S| + \sigma^2 \sum_{k \notin S} p_k - D, \tag{48}$$

where we have used cyclic property of trace, orthonormality of $\{\mathbf{r}_k\}_{k=1}^{D}$ and the fact that $p_k \lambda_k = 1$. We now define

$$f(S, \sigma^2) = \log|\mathbf{\Sigma}_p| - \sum_{k \in S} \log \lambda_k - (D - |S|) \log \sigma^2 + \sigma^2 \sum_{k \notin S} \lambda_k^{-1} + |S| - D, \tag{49}$$

and calculate

$$\frac{\partial f(S, \sigma^2)}{\partial \sigma^2} = \sum_{k \notin S} \lambda_k^{-1} - (D - |S|) \frac{1}{\sigma^2}. \tag{50}$$

Solving for an extremum gives $\sigma^2 = (\frac{1}{D-|S|} \sum_{k \notin S} \lambda_k^{-1})^{-1}$ being the harmonic mean of not selected eigenvalues. We substitute this $\sigma^2$ into the definition of $f(S, \sigma^2)$ to get

$$f(S) = \log |\mathbf{\Sigma}_p| - \sum_{k \in S} \log \lambda_k - (D - |S|) \log(\frac{1}{D - |S|} \sum_{k \notin S} \lambda_k^{-1})^{-1} + |S| + (D - |S|) - D. \tag{51}$$

Note that $\log |\mathbf{\Sigma}_p| = \sum_{k \in S} \log \lambda_k + \sum_{k \notin S} \log \lambda_k$ so that $\log |\mathbf{\Sigma}_p| - \sum_{k \in S} \log \lambda_k = \sum_{k \notin S} \log \lambda_k$. Thus we further have that

$$f(S) = \sum_{k \notin S} \log \lambda_k - (D - |S|) \log(\frac{1}{D - |S|} \sum_{k \notin S} \lambda_k^{-1})^{-1}. \tag{52}$$

We apply further transformations to $f(S)$ to get

$$f(S) = \sum_{k \notin S} \log \lambda_k - (D - |S|) \log(\frac{1}{D - |S|} \sum_{k \notin S} \lambda_k^{-1})^{-1} \tag{53}$$

$$= -(D - |S|) \frac{1}{D - |S|} \sum_{k \notin S} \log \lambda_k^{-1} + (D - |S|) \log \frac{1}{D - |S|} \sum_{k \notin S} \lambda_k^{-1} \tag{54}$$

$$= (D - |S|) \Big[ \log \frac{1}{D - |S|} \sum_{k \notin S} \lambda_k^{-1} - \frac{1}{D - |S|} \sum_{k \notin S} \log \lambda_k^{-1} \Big]. \tag{55}$$

Lastly we substitute $\lambda_k^{-1} = p_k$ to obtain:

$$h(S) = (D - |S|) \Big[ \log \frac{1}{D - |S|} \sum_{k \notin S} p_k - \frac{1}{D - |S|} \sum_{k \notin S} \log p_k \Big]. \tag{56}$$

Note that the expression $\log \frac{1}{D-|S|} \sum_{k \notin S} p_k - \frac{1}{D-|S|} \sum_{k \notin S} \log p_k$ is positive based on Jensen's inequality. Now we need to optimize $f(S)$ w.r.t. the index set $S$. The solution is to set $S$ having indices of $K$ smallest eigenvalues $p_k$ of precision matrix $\Lambda_p$.

To see this note that additionally we have a requirement on selected $p_k$ that $\sigma^2 = \frac{D-|S|}{\sum_{k \notin S} p_k}$ and $\lambda_k = \frac{1}{p_k} > \sigma^2$ so that $p_k < \frac{\sum_{k \notin S} p_k}{D-|S|}$ meaning that the selected $p_k$ has to be less than average over not selected $p_k$. It also follows that the largest $p_k$ has to be discarded. Based on Lemma 6 we have that $D - K$ largest $p_k$ have to be discarded leaving indices of $K$ smallest $p_k$ being selected to set $S$. Since $p_k = \frac{1}{\lambda_k}$ $K$ indices of largest $\lambda_k$ have to be selected to set $S$.

$\square$

### A.3 Computational cost

Here present detailed comparison of computational costs and provide explanations.

| Algorithm | Time | Memory |
|---|---|---|
| MAP | $\mathcal{O}(N_{in} N_{out} |\mathcal{B}|)$ | $\mathcal{O}(N_{out} |\mathcal{B}|)$ |
| naive mean-field | $\mathcal{O}(N_{in} N_{out} |\mathcal{B}|)$ | $\mathcal{O}(N_{in} N_{out} |\mathcal{B}|)$ |
| mean-field (LRT) | $\mathcal{O}(2 N_{in} N_{out} |\mathcal{B}|)$ | $\mathcal{O}(2 N_{out} |\mathcal{B}|)$ |
| naive low-rank | $\mathcal{O}(N_{in}^3 N_{out}^3 + N_{in} N_{out} |\mathcal{B}|)$ | $\mathcal{O}(N_{in} N_{out} |\mathcal{B}|)$ |
| efficient low-rank | $\mathcal{O}(K^3 + (K+2) N_{in} N_{out} |\mathcal{B}|)$ | $\mathcal{O}((K+2) N_{out} |\mathcal{B}|)$ |
| full rank | $\mathcal{O}(N_{in}^3 N_{out}^3 + N_{in}^2 N_{out}^2 |\mathcal{B}|)$ | $\mathcal{O}(N_{out} N_{in} |\mathcal{B}|)$ |

Table 6: Computational cost to update $\boldsymbol{\lambda}_l$ per layer.

Note that the above computational cost refers to forward pass through the network and calculating complexity penalty for corresponding layer (similar analysis can be done for backpropagation and would yield the same costs). We also do not discuss biases separately as they can by augmented with observations by extending the inputs. We discuss the memory usage we need to incur while

constructing dynamical computational graph. There is additional cost to store the parameters, for instance $\mathcal{O}(N_{in}^2 N_{out}^2)$ in the case of Gaussian variational posterior with full covariance matrix.

First, MAP estimation needs to store layer's output size $N_{out}$ numbers for every element in batch $\mathcal{B}$ and during forward pass it multiplies every parameter through corresponding input to the layer $x_i$ giving time cost $N_{in}N_{out}$. This cost is multiplied by the number of elements in batch $|\mathcal{B}|$. There is additional time cost proportional to $N_{in}N_{out}$ to estimate e.g. $L_2$ regularization.

Next, naive mean-field needs to perform the same computational effort as MAP, but it additionally needs to sample and store $|\mathcal{B}|$ sampled weights of dimension $N_{in}N_{out}$. Sampling weights costs $N_{in}N_{out}|\mathcal{B}|$ in both memory and time (sampling through reparametrization).

Local Reparametrization Trick improves upon naive mean-field as it requires sampling noise of shape $|\mathcal{B}|N_{out}$. It requires two forward passes to obtain means and variances of preactivations.

Naive reparametrization for low-rank plus diagonal Gaussian variational posterior requires to store $|\mathcal{B}| \times N_{in}N_{out}$ samples and perform forward pass of cost $|\mathcal{B}| \times N_{in}N_{out}$. There is additional cost $N_{in}^3 N_{out}^3$ incoming from estimating log determinant for complexity penalty.

Efficient low-rank requires sampling noise of shape $K$ and performing $K$ additional forward passes. We additionally need to add the cost of performing Local Reparametrization Trick. Using Lemma 2 allows to reduce the cost of calculating log determinant in complexity penalty to $K^3$.

## A.4 Local reparametrization for convolutional neural networks

We analyze a convolutional layer. We adopt the notation from Supplementary Material A.2. We denote input channel as $C_{in}$ and output channel as $C_{out}$. We have that $w^{ij,C_{in},C_{out}} := \sqrt{\alpha} \sum_{k=1}^{K} \epsilon_k v^{k,ij,C_{in},C_{out}} + \sigma^{ij,C_{in},C_{out}} \epsilon^{ij,C_{in},C_{out}}$ where $i,j$ run over coordinates of kernel. For clarity we drop the forward pass associated with mean $\boldsymbol{\mu}$, i.e. consider $\boldsymbol{\theta} \sim \mathcal{N}(\boldsymbol{\theta}|\mathbf{0}, \alpha \sum_{k=1}^{K} \mathbf{v}_k \mathbf{v}_k^\top + \mathrm{diag}[\boldsymbol{\sigma}^2])$. We have that for the output $o_p^{C_{out}}$ associated with patch $p$ and output channel $C_{out}$:

$$o_p^{C_{out}} = \sum_{C_{in}} \sum_{(i,j)\in p} w^{ij,C_{in},C_{out}} x^{ij,C_{in}} \tag{57}$$

$$= \sum_{C_{in}} \sum_{(i,j)\in p} \left[ \sqrt{\alpha} \sum_{k=1}^{K} \epsilon_k v_k^{ij,C_{in},C_{out}} x^{ij,C_{in}} + \sigma^{ij,C_{in},C_{out}} \epsilon^{ij,C_{in},C_{out}} \right] x^{ij,C_{in}} \tag{58}$$

$$= \sqrt{\alpha} \sum_{C_{in}} \sum_{(i,j)\in p} \sum_{k=1}^{K} \epsilon_k v_k^{ij,C_{in},C_{out}} x^{ij,C_{in}} + \sum_{C_{in}} \sum_{(i,j)\in p} \sigma^{ij,C_{in},C_{out}} \epsilon^{ij,C_{in},C_{out}} x^{ij,C_{in}} \tag{59}$$

$$= \sqrt{\alpha} \sum_{k=1}^{K} \epsilon_k \sum_{C^{in}} \sum_{(i,j)\in p} v_k^{ij,C_{in},C_{out}} x^{ij,C_{in}} + \sum_{C_{in}} \sum_{(i,j)\in p} \sigma^{ij,C_{in},C_{out}} \epsilon^{ij,C_{in},C_{out}} x^{ij,C_{in}}. \tag{60}$$

So similarly as in the case of the linear layer we can write output $o$ as

$$\mathbf{o} = \sqrt{\alpha} \sum_{k=1}^{K} \epsilon_k \mathcal{F}_{\mathbf{v}_k}(\mathbf{x}) + \mathcal{F}_{\boldsymbol{\psi}}(\mathbf{x}). \tag{61}$$

where $\boldsymbol{\psi} \sim \mathcal{N}(\mathbf{0}, \mathrm{diag}[\boldsymbol{\sigma}^2])$. The difference between fully connected and concolutional layer is that $\mathcal{F}_{\boldsymbol{\psi}}(\mathbf{x})$ cannot be reparametrized as LRT given by Equation (2) cannot be applied to convolutional layer [46] (since outputs for different patches $p$ are not independent due to sharing weights). Applying LRT to $\mathcal{F}_{\boldsymbol{\psi}}(\mathbf{x})$ can be treated as a rough approximation, but we did not find it to cause divergence.

## A.5 Experimental details

For toy experiment summarized in Figure 1 we run optimization for 30000 steps with learning rate 0.001 using ADAM optimizer. We initialize the $\mathrm{diag}[\boldsymbol{\sigma}^2]$ with $e^{-12}$ We use 200 data points for an update of parameters $\boldsymbol{\lambda}$ (whole data set $\mathcal{D}$). We use 500 variational samples to estimate statistics. We average the results over 30 random initializations and report corresponding error bars.

For vectorized MNIST classificaation with MLP summarized in Table 2 we estimate test statistics using 1000 variational samples. We do not employ data augmentation in this experiment. We normalize data set using empirical mean and standard deviation. We intitialize the layers mean sampling from $\mathcal{N}(0, \sqrt{\frac{2}{N_{in}+N_{out}}})$. We initialize the diag$[\boldsymbol{\sigma}^2]$ with $e^{-12}$ and initialize vectors $\mathbf{v}$ by sampling from $\mathcal{N}(0, 0.5\sqrt{\frac{2}{N_{in}+N_{out}}})$. We average the results over 10 random initializations and report corresponding error bars.

For LeNet classification experiment summarized in Table 3 we run optimization for 500 epochs (MAP 50) using batch size of 512, ADAM optimizer and learning rate 0.001 and one variational sample. For LeNet architecture we set log prior variance of units to $-5$. We use 500 variational samples to estimate test statistics. We normalize data sets using empirical mean and standard deviation. We do not employ data augmentation in this experiment. We average the results over 5 random initializations and report corresponding error bars.

For experiments with modern CNNs summarized in Table 4 we run optimization for 200 epochs (50 more than [35]). We run optimization using ADAM optimizer with learning rate 0.001 for first 100 epochs, then 0.0003 for remaining 100 epochs. We use one variational sample during training. We set log prior variance of units to $-1$ and initialize diag$[\boldsymbol{\sigma}^2]$ with $e^{-20}$ . We normalize data sets using empirical mean and standard deviation and employ data augmentation for these experiments: random padding followed by flipping left/right (except SVHN). We average the results over 5 random initializations and report corresponding error bars.