[Reviews · NeurIPS 2020]

Review 1

Summary and Contributions: The paper focuses on variational inference fo Bayesian neural networks with gaussian posteriors. The authors aim to improve the expressiveness of the approximate posterior and propose a memory-efficient low-rank approximation to the off-diagonal elements in the full covariance matrix. Furthermore, they propose a novel efficient local reparametrization trick to this more flexible posterior. Surprisingly, they find that augmenting a mean-field posterior does not seem to improve performance, but by taking a rank-0 approximation of the diagonal term, the authors do see improvement, validating the contribution.

Strengths: - Strongly positioned in related work, acknowledging both sides of the discussion for flexible or more restricted posteriors in variational Bayes. - To the best of my knowledge, the paper makes fundamental contributions in the derivation of the low-rank approximation of the off-diagonal term of the covariance matrix, as well as the local reparametrization trick. - Insightful experiments that acknowledge shortcomings and strengths.

Weaknesses: - The work would improve by providing better (visual) intuitions for what the proposed approximate posterior looks like and behaves like compared to a mean-field posterior. - The ELRG-VI posterior is in some sense more restricting than a mean-field posterior. This warrants empirical comparison to other papers that propose more-restricted approximate posteriors. In fact, the constant diagonal covariance can be considered the limiting case of the low-rank diagonal posterior by [Swiatkowski et al.] This suggests that such works should be incorporated in the empirical validations to explore how these two restricted posteriors compare.

Correctness: The proposed reparametrization trick and posterior appear correct.

Clarity: The paper is well written and is a joy to read.

Relation to Prior Work: The paper is well-positioned in both recent and older work in this area. The work clearly acknowledges ideas explored in related work and distinguish those from their contributions. Current areas of debate are highlighted appropriately.

Reproducibility: Yes

Additional Feedback: - [Swiatkowski et al.] https://arxiv.org/abs/2002.02655 Edit: Although the rebuttal addressed the concern regarding the comparison with the K-tied normal, a closer look at the large scale experiments in the rebuttal paint the proposed posterior in a less flattering perspective than the introduction and conclusion portray. I have decided to lower the score (partly due to my own calibration on reviewer scores), but still vote in favour of accept.


Review 2

Summary and Contributions: This paper proposes a reparametrization trick to speed up the computation for variational inference (VI) applied to neural networks, where the approximate posterior is a Gaussian family whose covariance is given by a sum of a low rank matrix and a diagonal matrix. By the reparametrization for the low rank structure in layer-wise forward pass, the computational cost is effectively reduced from cubically depending on the number of input and output neurons to linear dependence, which makes its application to large-scale neural networks tractable.

Strengths: The strengths of the paper are: 1. The proposed reparametrization can help to efficiently compute the update of the parameters per layer, which is scalable for large-scale deep neural networks. 2. The computational cost analysis is clear.

Weaknesses: The weaknesses are : 1. It is not clear how the reparametrization differ from the original parametrization with low rank covariance in terms of predictive performance. 2. It is not clear why the more expressive diagonal covariance is less predictive than the scaled identity covariance. 3. Why the latter covariance is more computationally efficient than the former for deep neural networks? 4. Given that the ELRG-VI has worse accuracy as in the Modern CNNs, what is the advantage of using this approximate posterior? 5. Evaluation of the quality of the posterior approximation by predictive performance is not appropriate as in many cases MAP can give more accurate predictions.

Correctness: The claims and method in terms of computational efficiency seems to be correct.

Clarity: The paper is a bit hard to follow with many notations not first defined and explained. For example, 1. what is p(y^*|x^*, \theta) in line 46? 2. what is the parametrization function g in line 59? 3. what is |B| in line 70?

Relation to Prior Work: The differences between this work and prior work are discussed. I am not sure if this work is significantly novel or original.

Reproducibility: Yes

Additional Feedback: I did not check the details of the setup of the empirical study and cannot say for sure about the reproducibility.


Review 3

Summary and Contributions: The authors explore the effects and quality of a variational approximation to the posterior of NNs based on low-rank Gaussian distributions per layer. In the course of this, they make the following contribunions: 1. exploration of a local reparametrization trick beyond diagonal Gaussians 2. the authors explore both low rank Gaussians as well as fixed diagonal and learned off-diagonal approximations. 3. the authors rigorously compare their two approximations, MFVI and MAP in a plethora of experiments and find that low rank MVN only performs well on small networks and their second approximation can give performance gains also on deeper architectures. I also want to point out that the paper is well-written, offers plenty of clear and appealing figures and tables and is overall well-executed.

Strengths: The paper lives on its empirical explorations. I think the strongest suit is the insights and comparisons of the two low rank factorizations and the discovery that small networks also work with LR-MVN, but deeper networks work better with the more heuristic fixed diagonal model. Another strong experimental insight is the discovery that per-sample variational samples improve performance vis-a-vis per batch variational samples. It is a nice add-on that the authors develop the local parametrization trick further.

Weaknesses: There are three crucial weaknesses for this paper. First, the paper is conceptually quite thin and does not have many fresh ideas to offer. This is also ok, as the paper is positioned to be very empirically focused. Second, given the empirical focus of the paper, I do not understand why the authors do not provide comparisons with the closely related 'k-tied Gaussians' by Swiatkowski et al paper that they cite in the introduction. How can an empirical analysis be complete for material focused on low-rank MVNs without that comparison? The first complete version of that paper was publicly presented at a Neurips workshop in 2019, so there has been enough time to be aware of its results. I understand the papers have slightly different focal points in that the k-tied paper focuses of covariance structures induced by weight columns, but for an empirical paper I would love to see that effect. Third, in the out-of-distribution experiments, it is unclear to me why the authors highlight their method as better 'due to the balance between uncertainty and confidence'. Is that what we'd expect here? My expectation of OOD data is that classes we have not trained on should yield close to uniform predictions, so I disagree with the author assessment here. Maybe they can provide a gold standard experiment, say via HMC, which can decide that? I am not clear this experiment here makes supporting statements for their method as claimed.

Correctness: Yes, the paper is overall of good quality and I did not detect jarring flaws.

Clarity: The paper is very well written, with clear notation, good overall structure, and great presentation.

Relation to Prior Work: I find the discussion and comparison to 'k-tied' lacking, given its relevance here. I am also perplexed how much this paper ignores Monte Carlo baselines for BNNs in order to evaluate its posteriors.

Reproducibility: Yes

Additional Feedback: I would consider raising my score with a rigorous comparison to K-Tied. Unfortunately I feel this is warranted for this paper, as the focus is empirical and the paper needs more comparators from the literature. Other than that I found the ideas I highlighted around per-datapoint variational samples and the effects of model depth quite appealing and would be interested in seeing more papers throw in such insights. Edit: In light of the rebuttal, I am raising my score to 6 and thank the authors for incorporating my suggestions.


Review 4

Summary and Contributions: This paper proposed adding a low-rank term to the local reparameterization trick (LRT) for improving the quality of mean-field approximation and adding additional regularization. Experimental results demonstrate that the proposed method may lead to improved test accuracy than the mean-field approach based on LRT with a small computational overhead. In addition, the uncertainty calibration results look better than MAP.

Strengths: The proposed approach modifies the LRT to facilitate low-rank Gaussian variational approximation for neural networks with a large number of parameters. The complexity of the proposed reparameterization is controlled by the matrix determinant lemma. Both isotropic and non-isotropic Gaussian variance are considered. The experiment evaluations are insightful. For example, the under-fitting issue of mean-field approach is highlighted, perhaps a little bit surprisingly, increasing the number of parameters in the proposed approach does not necessary improve the under-fitting. Moreover, the pitfalls of sharing parameters and uncertainty calibration results of baseline methods are presented.

Weaknesses: Although experimental results show that the proposed method can achieve a good balance between improved test accuracy, ELBO, and test uncertainty, I am still not sure about the takeaways from the paper, and how to use between MAP, MF-VI, and ELRG in practice. The improvement of quality in variational approximation is evidenced by the decrease of ELBO vs. K. It seems MAP can lead to better test accuracy but its uncertainty calibration is bad. Even for K=1, ELRG-VI has better test accuracy than MF-VI. But from Figure 3, it is unclear whether ELRG is better than MF-VI in characterizing uncertainty. Does the improvement of quality in variational approximation translate into better uncertainty quantification? In addition, I am not sure why low-rank plus isotropic variance (ELRG-VI) has better accuracy than the diagonal variant (ELRG-D-VI). Figure 2 (a) shows that for ELRG-D-VI, the negative log likelihoods are equivalent to approaches without low-rank terms. How to interpret the sharp drop and bump? Do the curves in Figure 3 (b) converge?

Correctness: The proposed approach seems correct except for the questions on the algorithm convergence.

Clarity: This paper is well-written and organized.

Relation to Prior Work: Yes.

Reproducibility: Yes

Additional Feedback: ###### Thank you for the rebuttal! I think the empirical study in this paper is interesting and therefore I increased my score to 6. I hope the following points get addressed in the updated version of this paper. (1) The underfitting of MF-VI is evidenced in Figure 2(c) and Figure 2(a) suggests increasing K in ELRG-D-VI doesn't help. While in the paper it is claimed that "ELRG-VI improves predictive performance WITHOUT underfitting", in Figure 2(b) the likelihood still decreases after ~10 epochs, just now increasing K makes a difference. In the rebuttal, the authors explain that the isotropic variance can force the posterior variances NOT close to prior variances, therefore migrating this issue. I read A.7 but still confused why. More explanations are needed. (2) As indicated by predictive entropy (Figure 4), the uncertainty level of the proposed approach sits in between MAP and MF-VI. But it seems all the methods are more uncertain on OOD data than test data. Based on such results, do we have enough reason to believe the ELRG-VI has a better uncertainty estimate than MF-VI? Is there any other evidence? Does it seem the ECE score results of MF-VI in Table 5 (appendix) is actually better?

[Author Response · NeurIPS 2020]

1  We thank the reviewers for their feedback and their time. (References here point to the submitted file's bibliography.)

2  **Novelty of the work.** We believe that R2 and R3 omitted key contributions that previous work has been unable to achieve: (i) our local reparameterization allows us to scale correlated Gaussian posteriors beyond what was thought to be possible previously, (ii) we show that a variational posterior 'LR + isotropic diag' outperforms existing VI methods for BNNs in most experiments, (iii) we show that correlated Gaussian posteriors offer a systematic solution to the underfitting of MF-VI, but this is limited only to small networks due to computational constraints, (iv) we identify problems with algorithms derived in previously published works [29, 32, 39, 41]. As R1 says, we believe these contributions to be significant for the community, and hope they will catalyze research on reparametrizing BNNs.

9  **R2: Predictive performance benefit of ELRG-VI vs original low-rank parametrization.** The original, naive, low-rank parametrization [41, 32, 39] is computationally expensive. This then requires sharing variational samples among inputs, leading to worse predictive performance (see Fig 4 (left)). Other benefits of separate variational samples are extensively studied in the literature (see references in Section 2) this was the primary motivation for work [19, 43].

13  **R2: "MAP can give more accurate predictions".** Although this is true, our results show that MAP has bad uncertainty calibration (as R4 says). We show problems with MAP estimation with calibration curves (Figure 3) and an out-of-distribution experiment. Losing slightly sometimes in accuracy but gaining a much better predictive distribution is often key for strong performance on real world tasks [4, 17]. Specifically, [11] shows overfitting to data yields better accuracy.

17  **R2 and R4: Why does ELRG-VI have a better predictive performance than ELRG-D-VI, with a more expressive non-isotropic diagonal?** We believe this confusing result arises due to a poor choice of prior over weights (see "Bayesian Deep Learning and a Probabilistic Perspective of Generalization", section 9.1), which becomes more apparent for large NNs. The general unintuitive problem that more expressive posteriors can yield worse predictions is an open research problem. For MF-VI, many units' variances converge to values close to prior variance, as shown in Figure 1 (densities), which is causing underfitting (see Appendix A7 for reasons). Adding low-rank (LR) terms mitigates this, but only for small networks. For larger networks, as K is small compared to matrix size, the described mechanism is negligible, and we need to resort to using a simpler posterior where units cannot converge to the prior. R2 also asked why LR + isotropic diagonal is a better choice for specifically CNNs. As per line 130, for CNNs, LRT cannot be applied [43], but the LR component can still be reparametrized. In practice, many authors still use LRT for CNNs, claiming this is an approximation. Using LR + isotropic diagonal with a tiny diagonal scale reduces the effect of this approximation as the problematic diagonal term is very small (see Appendix A.3). We will improve this writing in the paper.

29  **R3 (and R1): Comparison to Swiatkowski et al., 2020, "k-tied".** We provide a comparison on a toy experiment here (see Figure) and will provide a more extensive comparison for the camera-ready version. However, we feel one should be careful about excessively focussing on the conclusions from such a comparison. K-tied approach lowers the complexity of the posterior, whereas our paper adds low-rank terms to increase the expressiveness. In theory our approach should yield better predictions than k-tied if ELBO is correlated with predictive performance. If this is not happening, it is likely we face issues with problem formulation, possibly due to a poor prior distribution (see discussion above for details). In terms of ELBO values, ELRG-D-VI $\geq$ MF-VI $\geq$ k-tied, assuming we find the global maximizer. As R1 says, it is hard to compare with ELRG-VI's ELBO value. We also note that k-tied + our approach will yield higher ELBO than k-tied alone. Here, we test on the same setup for Figure 1 in the main paper. We find that k-tied results in higher ELBO than ELRG-VI (ELRG-VI value worse by a factor of 10 than other algorithms, hence not plotted), but ELRG-VI achieves the best predictive performance. For small NNs, we observe that k-tied performance matches standard MF-VI and does not change with K. Also, we compare (Figure 2b) to another work [42] by increasing the complexity of WN posterior discussed there.

41  **R3: Questions about the out-of-distribution (OOD) experiment.**
43  R3 has misunderstood the context of this experiment. We specifically refer to confidence for **test** and **train** data (not OOD data) corresponding to the dataset used to train the model. Lack of confidence for test/train data means poor predictive performance (such as with MF-VI). As R3 notes, we then expect high uncertainty for OOD data. MF-VI gives high uncertainty both on test data and OOD data but offers weak predictions due to underfitting. We will clarify

50  these points in the text. The OOD experiment closely follows the related paper [33] and matches their conclusions.

51  **R4: ELRG-VI vs MF-VI predictive uncertainty.** From our experiments, it is visible MF-VI yields poor predictions and predictive uncertainty compared to ELRG-VI (Table 3 and Fig 4). We will improve the discussion of this point.

53  **R4: Shape and convergence of curves in Figure 2.** In Fig 2, the experiments for $\approx$400K optimization steps. This number is a highly excessive number of updates for MNIST, many more than other papers ($\approx$100K more than Bayes by Backprop). Learning curves will change insignificantly if optimization is run for longer. R4 asked about the shape of curves in Figure 2 (a,b). The momentum in the ADAM optimizer causes the bump (please note the curves show *validation* performance, not the optimized quantity). R4 also asked about the equivalence of approaches without LR in Figure 2(a). It turns out that isotropic diagonal has slightly better performance, in line with conclusions in [42].

[Meta-Review · NeurIPS 2020]

The authors explore the effects and quality of a variational approximation to the posterior of NNs based on low-rank Gaussian distributions per layer. Strengths: - the paper is well written and covers the literature well - the novel local reparameterization trick and low-rank variational approximation are sound and efficient Weaknesses: - missing comparisons to baselines (these were promised to be delivered upon acceptance)